# UniEdit: A Unified Tuning-Free Framework for Video Motion and Appearance Editing

Project webpage: https://uni-edit.github.io/UniEdit/

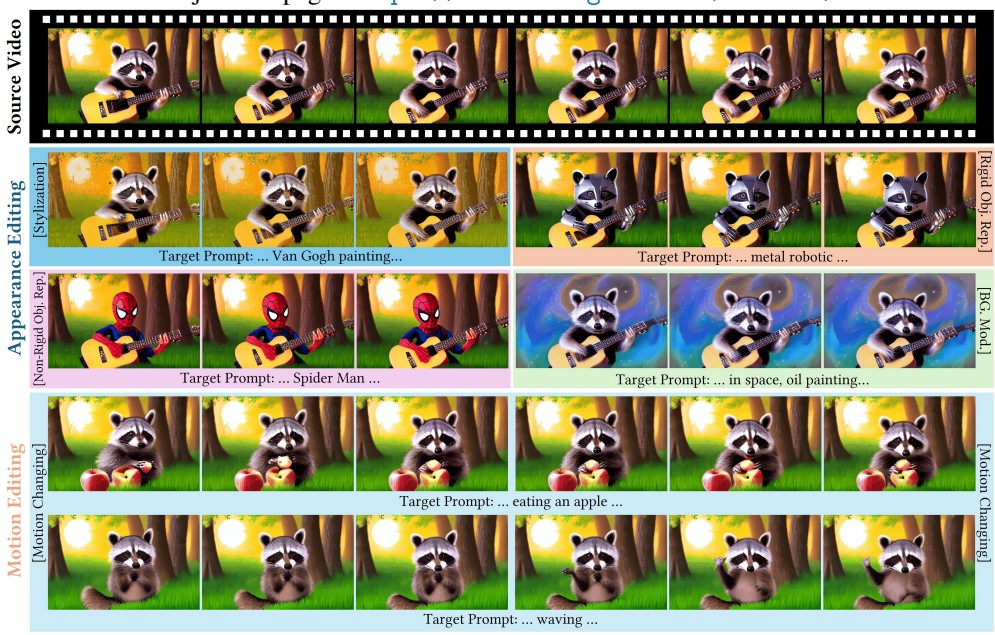

Figure 1: Examples edited by UniEdit. Our solution supports both video *motion* editing in the time axis (i.e., from playing guitar to eating or waving) and various video *appearance* editing scenarios (i.e., stylization, rigid/non-rigid object replacement, background modification). We encourage the readers to watch the videos on our project page.

## Abstract

Recent advances in text-guided video editing have showcased promising results in appearance editing (e.g., stylization). However, video motion editing in the temporal dimension (e.g., from eating to waving), which distinguishes video editing from image editing, is underexplored. In this work, we present UniEdit, a tuning-free framework that supports both video motion and appearance editing by harnessing the power of a pre-trained text-to-video generator within an inversion-then-generation framework. To realize motion editing while preserving source video content, based on the insights that temporal and spatial self-attention layers encode inter-frame and intra-frame dependency respectively, we introduce auxiliary motion-reference and reconstruction branches to produce text-guided motion and source features respectively. The obtained features are then injected into the main editing path via temporal and spatial self-attention layers. Extensive experiments demonstrate that UniEdit covers video motion editing and various appearance editing scenarios, and surpasses the state-of-the-art methods. Our code will be publicly available.

Submitted to 38th Conference on Neural Information Processing Systems (NeurIPS 2024). Do not distribute.

# 1 Introduction

The advent of pre-trained diffusion-based [26, 53] text-to-image generators [49, 50, 48] has revolutionized the fields of design and filmmaking, opening new vistas for creative expression. These advancements, underpinned by seminal works in text-to-image synthesis, have paved the way for innovative text-guided editing techniques for both images [42, 24, 4, 5] and videos [65, 6, 39, 70, 17, 46]. Such techniques not only enhance creative workflows but also promise to redefine content creation within these industries.

Video editing, in contrast to image editing, introduces the intricate challenge of ensuring frame-wise consistency. Efforts to address this challenge have led to the development of methods that leverage shared features and structures with the source video [6, 39, 37, 70, 46, 7, 33, 62, 18] through an inversion-then-generation pipeline [42, 53], exemplified by Pix2Video's approach [6] to consistent appearance editing across frames. To transfer the edited appearance from the anchor frame to the remaining frames consistently, it employs a pre-trained image generator and extends the self-attention layers to cross-frame attention to generate each remaining frame. Despite these advancements in performing video *appearance* editing (e.g., stylization, object appearance replacement, etc.), these methodologies fall short in editing video *motion* (e.g., replacing the movement of playing guitar with waving), hampered by a lack of motion priors and limited control over inter-frame dependencies, underscoring a critical gap in video editing capabilities.

Previous attempts [65, 44] at video motion editing through fine-tuning a pre-trained generator on the given source video and then editing motion through text guidance. Although effective, they necessitate a delicate balance between the generative prowess of the model and the preservation of the source video's content. This compromise often leads to restricted motion diversity and unwanted content variations, indicating a pressing need for a more robust solution.

In response, our work aims to explore a *tuning-free* framework that adeptly navigates the complexities of editing both the *motion* and *appearance* of videos. To achieve this, we identify three technical challenges: 1) it is non-trivial to incorporate the text-guided motion into the source content, as directly applying video appearance editing [46, 18] or image editing [5] schemes leads to undesirable results (as shown in Fig. 5); 2) preserving the non-edited content of the source video; 3) inheriting the spatial structure of the source video during appearance editing.

Our solution, UniEdit, harnesses the power of a pre-trained text-to-video generator (e.g., LaVie [63]) within an inversion-then-generation framework [42], tailored to overcome the identified challenges. Particularly, we introduce three key innovations: 1) To inject text-guided motion into the source content, we highlight the insight that *the temporal self-attention layers of the generator encode the inter-frame dependency*. Acting in this way, we introduce an auxiliary motion-reference branch to generate text-guided motion features, which are then injected into the main editing path via temporal self-attention layers. 2) To preserve the non-edited content of the source video, motivated by the image editing technique [5], we follow the insight that *the spatial self-attention layers of the generator encode the intra-frame dependency*. Therefore, we introduce an auxiliary reconstruction branch, and inject the features obtained from the spatial self-attention layers of the reconstruction branch into the main editing path. 3) To retain the spatial structure during the appearance editing, we replace the spatial attention maps of the main editing path with those in the reconstruction branch.

To our best knowledge, UniEdit represents a pioneering leap in text-guided, tuning-free video motion editing. In addition, its unified architecture not only facilitates a wide array of video appearance editing tasks, as shown in Fig. 1, but also empowers image-to-video generators for zero-shot text-image-to-video generation. Through comprehensive experimentation, we demonstrate UniEdit's superior performance relative to existing state-of-the-art methods, highlighting its potential to significantly advance the field of video editing.

# 2 Related Works

## 2.1 Video Generation

Researchers have achieved video generation with generative adversarial networks [58, 51, 61], language models [69, 71], or diffusion models [28, 52, 25, 23, 3, 60, 72, 19, 63, 8, 47]. To make the generation more controllable, recent endeavors have also incorporated additional structure guidance (e.g., depth map) [16, 10, 74, 11, 20, 64], or conducted customized generation [65, 67, 34, 75, 59, 41].

These models have generally learned real-world video distribution from large-scale data, and achieved promising results on text-to-video or image-to-video generation. Based on their success, we leverage the learned prior in the pre-trained model to achieve tuning-free video motion and appearance editing.

## 2.2 Video Editing

Video editing aims to produce a new video that is aligned with the provided editing instructions (e.g., text) while maintaining the other characteristics of the source video. It can be categorized into appearance and motion editing.

For appearance editing [70, 15, 17, 35, 12], like turn a video into the style of Van Gogh, the main challenge is to achieve temporal-consistent generation across different frames. Early attempts [6, 37, 46, 7, 33, 62] leveraged text-to-image models with inter-frame propagation to ensure consistency. For instance, Pix2Video [6] replaces the key and value of the current frame with those of the first and previous frame. Video-P2P [39] achieved local editing via video-specific fine-tuning and unconditional embedding optimization [43]. Follow-up studies [18, 70, 45] also leveraged the edit-then-propagate framework with neatest-neighbor field [18], estimated optical flow [70], or temporal deformation field [45]. Despite the promising results, due to the constraint on the source video structure, these approaches are specialized in appearance editing and can not be applied to motion editing directly.

Recent studies have also explored video motion editing with text guidance [65, 44], user-provided motion [32, 54, 15], or specific motion representation [55, 36, 22]. For example, Dreamix [44] proposed fine-tuning a pre-trained text-to-video model with mixed video-image reconstruction objectives for each source video. Then the editing is realized by conditioning the fine-tuned model on the given target prompt. MoCA [68] decoupled the video into the first-frame appearance and the optical flow, and trained a diffusion model to generate video conditioned on the first frame and the text. However, it struggled to preserve the non-edited motion (e.g., background dynamics) as it generates the entire motion from the text. Different from the aforementioned approaches that require fine-tuning or user-provided motion input, we are the first to achieve tuning-free motion and appearance editing with text guidance only.

# 3 Preliminaries: Video Diffusion Models

Our proposed UniEdit is built upon video diffusion models. Therefore, we first recap the architecture that is used in common text-guided video diffusion models [63, 2].

**Overall Architecture** Modern text-to-video (T2V) diffusion models typically extend a pre-trained text-to-image (T2I) model [49] to the video domain with the following adaptations. 1) Introducing additional temporal layers by inflating 2d convolutional layers to 3d form, or adding temporal self-attention layers [57] to model the correlation between video frames. 2) Due to the extensive computational resources for modeling spatial-temporal joint distribution, these works typically first train video generation models on low spatial and temporal resolutions, and then upsampling the generated results with cascaded models. 3) Other improvements like efficiency [1], training strategy [19], or additional control signals [16], etc. During inference, given standard Gaussian distribution $z_T \sim \mathcal{N}(0, 1)$, the denoising UNet is used to perform $T$ denoising steps to obtain the outputs [26, 53]. If the model is trained in latent space [49], a decoder is employed to reconstruct videos from the latent domain.

**Attention Mechanisms** In particular, for each block of the denoising UNet, there are four basic modules: a convolutional module, a spatial self-attention module (SA-S), a spatial cross-attention module (CA-S), and a temporal self-attention module (SA-T). Formally, the attention operation [57] can be formulated as:

$$\mathtt{attn}(Q, K, V) = \mathtt{softmax}(\frac{QK^T}{\sqrt{d}})V, \tag{1}$$

where $Q$ (query), $K$ (key), $V$ (value) are derived from inputs, and $d$ is the dimension of hidden states.

Intuitively, CA-S is in charge of fusing semantics from the text condition, SA-S models the intra-frame dependency, SA-T models the inter-frame dependency and ensures the generated results are temporally consistent. We leverage these intuitions in our designs as elaborated below.

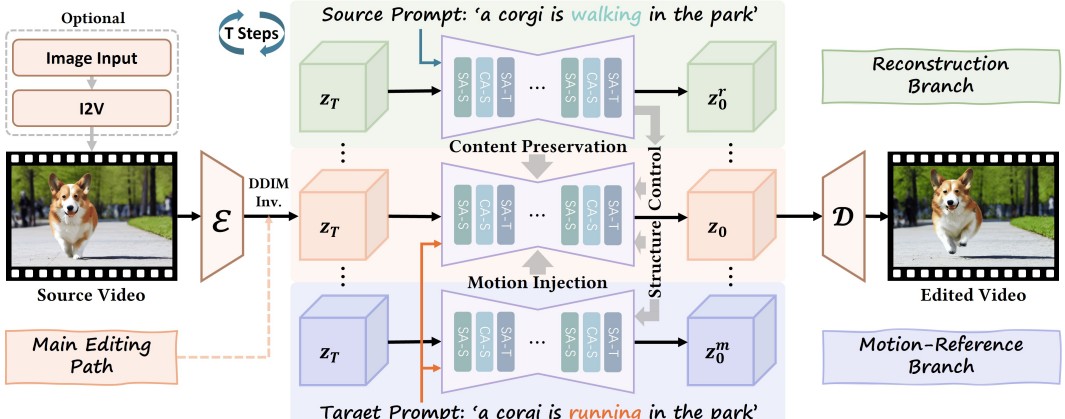

Figure 2: Overview of UniEdit. It follows an inversion-then-generation pipeline and consists of a main editing path, an auxiliary reconstruction branch and an auxiliary motion-reference branch. The reconstruction branch produces source features for content preservation, and the motion-reference branch yields text-guided motion features for motion injection. The source features and motion features are injected into the main editing path through spatial self-attention (SA-S) and temporal self-attention (SA-T) modules respectively (Sec. 4.1). We further introduce spatial structure control to retain the coarse structure of the source video (Sec. 4.2).

## 4 UniEdit

**Method Overview.** As shown in Fig. 2, our main editing path is based on an inversion-then-generation pipeline: we use the latent after DDIM inversion [53] as the initial noise $z_T$[1], then perform denoising process starting from $z_T$ with the pre-trained UNet conditioned on the target prompt $P_t$. For motion editing, to achieve source content preservation and motion control, we propose to incorporate an auxiliary reconstruction branch and an auxiliary motion-reference branch to provide desired source and motion features, which are injected into the main editing path to achieve content preservation and motion editing (as shown in Fig. 3). We propose the pipeline of motion editing and appearance editing in Sec. 4.1 & Sec. 4.2 respectively. To further alleviate the background inconsistency, we introduce a mask-guided coordination scheme in Sec. 4.3. We also extend UniEdit to text-image-to-video generation (TI2V) in Sec. 4.4.

### 4.1 Tuning-Free Video Motion Editing

**Content Preservation on SA-S Modules.** One of the key challenges of editing tasks is to inherit the original content (e.g., textures and background) in the source video. To this end, we introduce an auxiliary reconstruction branch. The reconstruction path starts from the same inversed latent $z_T$ similar to the main editing path, and then conducts the denoising process with the pre-trained UNet conditioned on the source prompt $P_s$ to reconstruct the original frames. As verified in image editing [56, 24, 5], the attention features in the denoising model during reconstruction contain the content of the source video. Hence, we inject attention features of the reconstruction path into the main editing path on spatial self-attention (SA-S) layers for content preservation. At denoising step $t$, the attention operation of the $l$-th SA-S module in the main editing path is formulated as:

$$\text{SA-S}_{\text{edit}}^l := \begin{cases} \texttt{attn}(Q, K, V^r), & t < t_0 \text{ and } l > L, \\ \texttt{attn}(Q, K, V), & \text{otherwise,} \end{cases} \tag{2}$$

where $Q, K, V$ are features in the main editing path, $V^r$ refer to the value feature of the corresponding SA-S layer in the reconstruction branch, $t_0 = 50$ and $L = 10$ are hyper-parameters following previous work [5]. By replacing the value of spatial features, the video synthesized by the main editing path retains the non-edited characters (e.g., identity and background) of the source video, as exhibited in Fig. 7a. Unlike previous video editing works [37, 29] which introduces a cross-frame attention mechanism (i.e., using the key and value of the first/last frame), we implement Eq. 2 frame-wisely to better tackle source video with large dynamics.

---

[1]For real source video, we set source prompt to null during both forward and inversion process to achieve high-quality reconstruction [43].

**Motion Injection on SA-T Modules.** After implementing the content-preserving technique introduced above, we can obtain an edited video with the same content in the source video. However, it is observed that the output video could not follow the text prompt $P_t$ properly. A straightforward solution is to increase the value of $L$ so that balancing between the impact of injected information and the conditioned text prompt. Nevertheless, this could result in a content mismatch with the original source video in terms of structures and textures.

To obtain the desired motion without sacrificing content consistency, we propose to guide the main editing path with reference motion. Concretely, an auxiliary motion-reference branch (which also starts from the inversed latent $z_T$) is involved during the denoising process. Different from the reconstruction branch, the motion-reference branch is conditioned on the target prompt $P_t$, which contains the description of the desired motion. To transfer the motion into the main editing path, our core insight here is that ***temporal layers model the inter-frame dependency of the synthesized video clip*** (as shown in Fig. 6). Motivated by the observations above, we design the attention map injection on temporal self-attention layers of the main editing path:

$$\text{SA-T}^l_{\text{edit}} := \texttt{attn}(Q^m, K^m, V) \tag{3}$$

where $Q^m$ and $K^m$ refer to the query and key of the motion-reference branch, note that we replace the query and key of SA-T modules in the main editing path with those in the motion-reference branch on all layers and denoising steps. It's observed that the injection of temporal attention maps can effectively facilitate the main editing path to generate motion aligned with the target prompt. To better fuse the motion with the content in the source video, we also implement spatial structure control (refer to Sec. 4.2) on the main editing path and motion-reference branch in the early steps.

## 4.2 Tuning-Free Video Appearance Editing

In Sec. 4.1, we introduce the pipeline of UniEdit for video motion editing. In this subsection, we aim to perform appearance editing (e.g., style transfer, object replacement, background changing) via the same framework. In general, there are two main differences between appearance editing and motion editing. Firstly, appearance editing does not require changing the inter-frame relationships. Therefore, we remove the motion-reference branch and corresponding motion injection mechanism from the motion editing pipeline. Secondly, the main challenge of appearance editing is to maintain the structural consistency of the source video. To address this, we introduce spatial structure control between the main editing path and the reconstruction branch.

Figure 3: Detailed illustration of the relationship between the main editing path, the auxiliary reconstruction branch and the auxiliary motion-reference branch. The content preservation, motion injection and spatial structure control are achieved by the fusion of $Q$ (query), $K$ (key), $V$ (value) features in spatial self-attention (SA-S) and temporal self-attention (SA-T) modules.

**Spatial Structure Control on SA-S Modules.**
Previous approaches on video appearance editing [70, 18] mainly realize spatial structure control with the assistance of additional network [73]. When the auxiliary control model fails, it may result in inferior performance in preserving the structure of the original video. Alternatively, we suggest extracting the layout information of the source video from the reconstruction branch. Intuitively, the attention maps in spatial self-attention layers encode the structure of the synthesized video, as verified in Fig. 6. Hence, we replace the query and key of SA-S module in the main editing path with those in the reconstruction branch:

$$\text{SA-S}^l_{\text{edit}} := \begin{cases} \texttt{attn}(Q^r, K^r, V), & t < t_1, \\ \texttt{attn}(Q, K, V), & \text{otherwise,} \end{cases} \tag{4}$$

where $Q^r$ and $K^r$ refer to the query and key of the reconstruction branch, $t_1$ is used to control the extent of editing. It is worth mentioning that the effect of spatial structure control is distinct from the content preservation mechanism in Sec. 4.1. Take stylization as an example, the proposed structure control in Eq. 4 only ensures consistency in terms of each frame's composition, while enabling the model to generate the required textures and styles based on the text prompt. On the other hand,

the content preservation technique inherits the textures and style of the source video. Therefore, we use structure control instead of content preservation for appearance editing. In addition, using the proposed structure control technique in motion editing can make the layout of the output video similar to the source video (shown in Fig. 11b in Appendix). Users have the flexibility to adjust the consistency between the edited video and the source video layout based on their specific requirements.

### 4.3 Mask-Guided Coordination (Optional)

To further improve the editing performance, we suggest leveraging the foreground/background segmentation mask $M$ to guide the denoising process [14, 13]. There are two possible ways to obtain the mask $M$: the attention maps of CA-S modules with a threshold [24]; or employing an off-the-shelf segmentation model [38] on the source and generated videos. The obtained segmentation masks can be leveraged to 1), alleviate the indistinction in foreground and background; 2), improve content consistency between edited and source videos. To this end, we leverage mask-guided self-attention in the main editing path to coordinate the editing process. Formally, we define:

$$\texttt{m-attn}(Q, K, V; M) = \texttt{softmax}(\frac{QK^T}{\sqrt{d}} + M)V. \tag{5}$$

Then the mask-guided self-attention:

$$\text{SA}_{\text{mask}} := \texttt{m-attn}(Q, K, V; M^f) \odot M_m + \texttt{m-attn}(Q, K, V; M^b) \odot (1 - M_m), \tag{6}$$

where $M^f, M^b \in \{-\infty, 0\}$ indicate the foreground and background masks in the editing path respectively, $M_m \in \{0, 1\}$ denotes the foreground mask from the motion-reference branch, and $\odot$ is Hadamard product. In addition, we leverage the mask during the content preservation and motion injection for the features obtained from the reconstruction branch and the motion-reference branch (e.g., we replace $Q^m$ with $M_m \odot Q^m + (1 - M_m) \odot Q$).

### 4.4 T2V Models are Zero-Shot TI2V Generators

To make our framework more flexible, we further derive a method to incorporate images as input and synthesize high-quality video conditioned on *both* image and text-prompt. Different from some image animation techniques [2], our method allows the user to guide the animation process with text prompts. Concretely, we first achieve image-to-video (I2V) generation by: 1) transforming input images with simulated camera movement to form a pseudo-video clip [44] or 2) leveraging existing image animation approaches (e.g., SVD [2], AnimateDiff [21]) to synthesis a video with random motion (which may not consistent with the text prompt). Then, we perform text-guided editing with UniEdit on the vanilla video to obtain the final output video.

## 5 Experiments

### 5.1 Comparison with State-of-the-Art Methods

**Implementation Details** UniEdit is not limited to specific video diffusion models. In this section, we build UniEdit upon LaVie [63] as an instantiation to verify the effectiveness of our method. To demonstrate the flexibility of UniEdit across different base models, we also implement the proposed method on VideoCrafter2 [9] and exhibit the editing results in Appendix B.1. For each input video, we follow the pre-processing step in LaVie to the resolution of $320 \times 512$. Then, the pre-processed video is fed into the UniEdit to perform video editing. It takes 1-2 minutes to edit on an NVIDIA A100 GPU for each video. More details can be found in Appendix A.

**Baselines.** To evaluate the performance of UniEdit, we compare the editing results of UniEdit with state-of-the-art motion and appearance editing approaches. For motion editing, due to the lack of open-source tuning-free (zero-shot) methods, we adapt the state-of-the-art non-rigid image editing technique MasaCtrl [5] to a T2V model [63] (denoted as MasaCtrl* in Fig. 5) and a one-shot video editing method Tune-A-Video (TAV) [65] as strong baselines. For appearance editing, we use the latest methods with strong performance, including FateZero [46], TokenFlow [18], and Rerender-A-Video (Rerender) [70] as baselines.

**Evaluation Set.** The evaluation set consists of 100 samples, including: **a)** 20 randomly sampled video clips from the open-source LOVEU-TGVE-2023 [66] dataset, along with their corresponding 80 text prompts, and **b)** 20 videos from online sources (www.pexels.com and www.pixabay.com), with manually designed prompts, as the baseline methods do not have an open-source evaluation set.

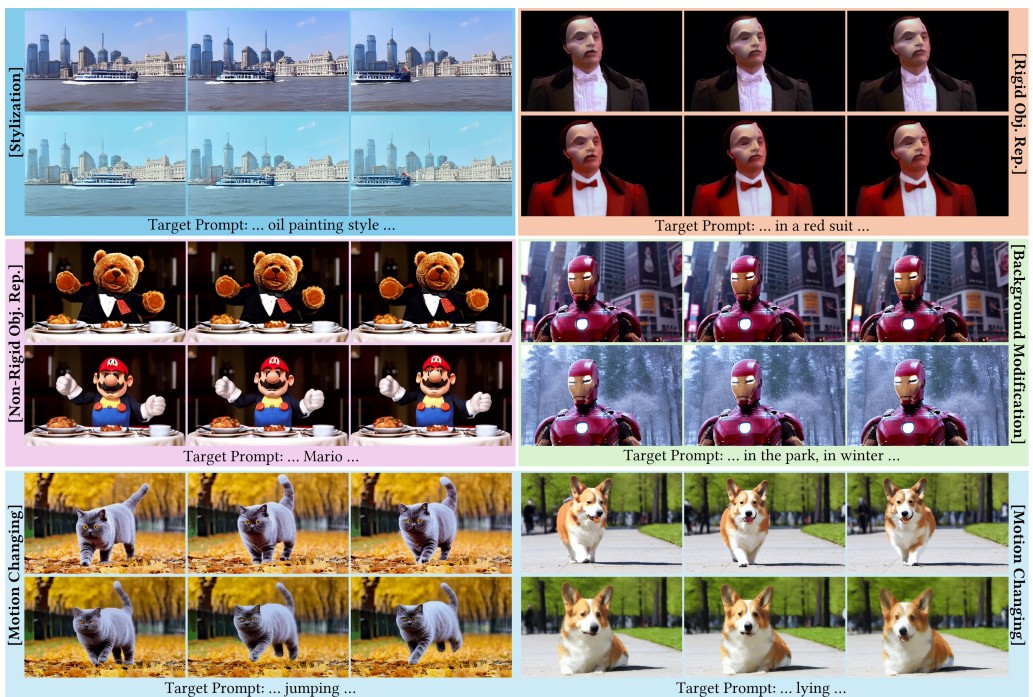

Figure 4: Examples edited by UniEdit. For each case, the upper frames come from the source video, and the lower frames indicate the edited results with the target prompt. We encourage the readers to watch the videos and make evaluations.

**Qualitative Results.** We present editing examples of UniEdit in Fig. 1, Fig. 4 (additional examples in Fig. 16-21 of Appendix B.8). Please visit our project page for more videos. UniEdit demonstrates the ability to: 1) edit in various scenarios, including motion-changing, object replacement, style transfer, and background modification; 2) align with the target prompt; and 3) maintain excellent temporal consistency. Additionally, we compare UniEdit with state-of-the-art methods in Fig. 5 (further comparisons in Fig.13,14,15 of Appendix B.7). For a fair comparison, we also migrated all baselines to LaVie [63], using the same base model as our method. The results are presented in Fig. 15. For appearance editing, we showcase two scenarios: non-rigid object replacement and stylization. In object replacement, our method outperforms baselines in terms of prompt alignment and background consistency. In stylization, UniEdit excels in preserving content. For example, the grassland retains its original appearance without any additional elements. In motion editing, UniEdit surpasses baselines in aligning the video with the target prompt and preserving the source content.

**Quantitative Results.** We quantitatively evaluate our method using two approaches: **1)** CLIP scores and user preference, as employed in previous work [65]; and **2)** VBench [31] scores, a recently proposed benchmark suite for T2V models. The summarized results are in Tab. 1. Following previous work [65], we assess the effectiveness of our method in terms of temporal consistency and alignment with the target prompt. Additionally, we conducted a user study involving 10 participants who rated the edited videos on a scale of 1 to 5. We also utilize the recently proposed VBench [31] benchmark to provide a more comprehensive assessment, which includes 'Frame Quality' metrics and 'Temporal Quality' metrics. UniEdit outperforms the baseline methods across all metrics. Furthermore, the mask-guided coordination technique introduced in Sec. 4.3 further enhances performance (see Appendix B.3). For more detailed quantitative results, please refer to Appendix B.2&B.3&B.5.

## 5.2 Ablation Study and Analysis

**How UniEdit Works?** To better understand how UniEdit works and reveal our insight on the spatial and temporal self-attention layers, we visualize the features in the SA-S and SA-T modules and compare them with the magnitude of optical flow between adjacent frames in Fig. 6a. It is evident that, in comparison to the spatial query maps (2nd row), the temporal cross-frame attention maps (3rd row) exhibit a notably higher degree of overlap with the optical flow (4th row). This indicates that the temporal self-attention layers encode inter-frame dependencies and facilitate motion injection, while content preservation and structure control are carried out in the spatial self-attention layers.

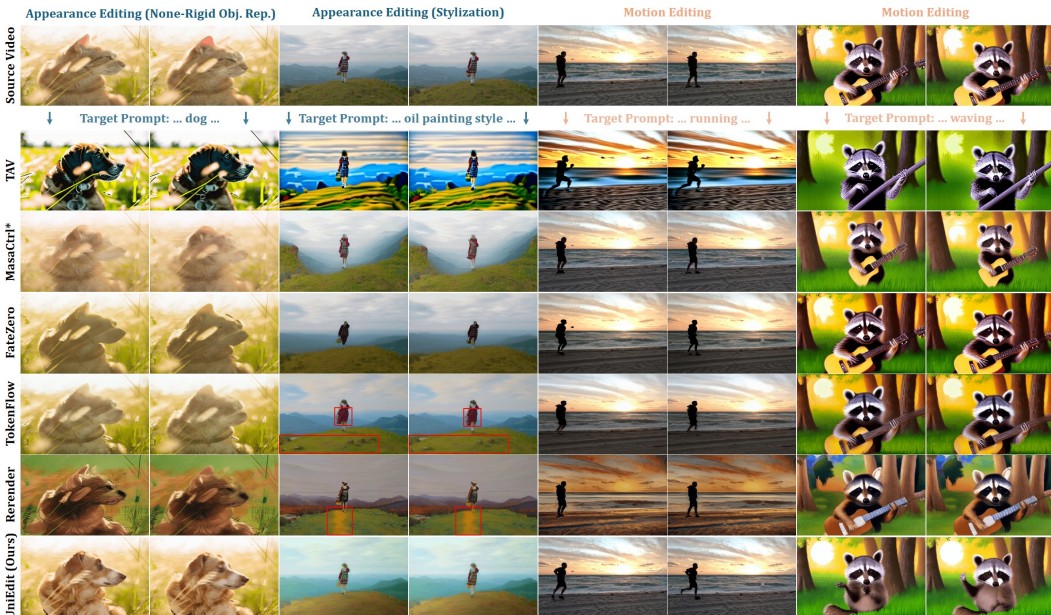

Figure 5: Comparison with state-of-the-art methods for both video appearance and motion editing. It shows that UniEdit achieves better source content preservation, and outperforms baselines in motion editing by a large margin.

Table 1: Quantitative comparison with state-of-the-art video editing techniques.

| Method | Frame Consistency | | Textual Alignment | | Frame Quality | | Temporal Quality | | |
| | CLIP Score | User Pref. | CLIP Score | User Pref. | Aesthetic Quality | Imaging Quality | Subject Consistency | Motion Smoothness | Temporal Flickering |
|---|---|---|---|---|---|---|---|---|---|
| TAV [65] | 95.39 | 3.74 | 27.89 | 3.30 | 51.97 | 49.60 | 93.10 | 93.27 | 91.48 |
| MasaCtrl* [5] | 97.61 | 4.31 | 25.58 | 3.17 | 54.58 | 58.72 | 93.04 | 95.70 | 94.29 |
| FateZero [46] | 96.72 | 4.48 | 27.30 | 3.48 | 53.77 | 56.99 | 93.55 | 94.80 | 93.42 |
| Rerender [70] | 97.18 | 4.16 | 27.94 | 3.55 | 54.59 | 57.97 | 93.08 | 95.57 | 94.36 |
| TokenFlow[18] | 97.02 | 4.50 | 28.58 | 3.34 | 52.60 | 60.65 | 91.97 | 95.04 | 93.50 |
| UniEdit | 98.35 | 4.72 | 31.43 | 4.79 | 58.25 | 62.94 | 95.73 | **97.30** | 96.74 |
| UniEdit-Mask | **98.36** | **4.73** | **31.50** | **4.90** | **58.77** | **63.12** | **95.86** | 97.28 | **96.79** |

**Output Visualization of the Two Auxiliary Branches.**  Recall that to perform motion editing, we propose to transfer the targeted motion from the motion-reference branch and realize content preservation via feature injection from the reconstruction branch. To verify the effectiveness, we visualized the output of each branch in Fig. 6b. It is observed that the motion-reference branch (4th row) generates video with the target motion, and effectively transfers it to the main path (3rd row); meanwhile, the main path inherits the content from the reconstruction branch (2nd row), thus enhancing the consistency of unedited parts.

**The Effectiveness of Each Component.**  To demonstrate that all the designed feature injection techniques in Sec. 4.1 & 4.2 contribute to the final results, we make a quantitative evaluation on 15 motion editing cases, as we utilize all three components in motion editing. To assess the similarity between the edited video and the source video (e.g., background and identity), we introduce the 'Frame Similarity', which is the average frame cosine similarity between the source frame embedding and

Table 2: Impact of various components.

| Content Preservation | Motion Injection | Structure Control | Frame Similarity | Textual Alignment | Frame Consistency |
|---|---|---|---|---|---|
| | | | 90.54 | 28.76 | 96.99 |
| ✓ | | | 97.28 | 29.95 | 98.12 |
| | ✓ | ✓ | 91.30 | 31.48 | 98.08 |
| ✓ | ✓ | | 96.11 | 31.37 | 98.12 |
| ✓ | ✓ | ✓ | 96.29 | 31.43 | 98.09 |

the edited frame embedding. As shown in Tab. 2, editing with *content preservation* results in high frame similarity, suggesting that replacing value features in SA-S modules can effectively retain the content of the source video. The use of *motion injection* and *structure control* significantly enhances 'Textual Alignment', indicating successful transfer of the targeted motion to the main editing path. Ultimately, the best results are achieved through the combined use of all components.

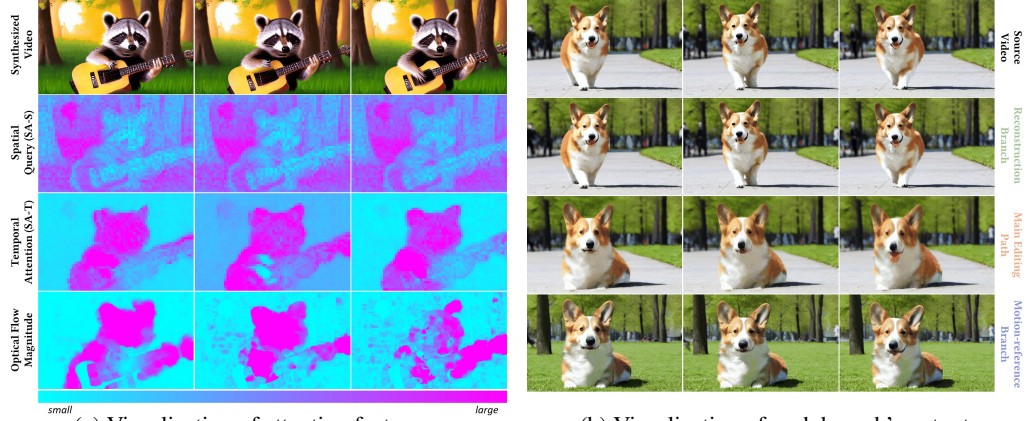

(a) Visualization of attention features.  (b) Visualization of each branch's output.

Figure 6: (6a): Visualization of spatial query in SA-S (second row), cross-frame temporal attention maps in SA-T (third row), and the magnitude of optical flow (fourth row). (6b): Visualization of the video output of the main editing path, the reconstruction branch and the motion-reference branch.

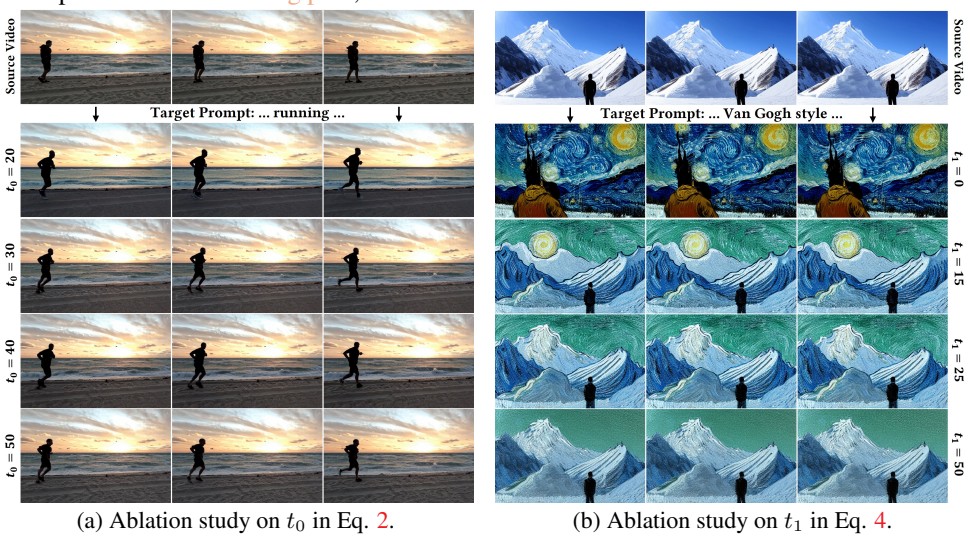

(a) Ablation study on $t_0$ in Eq. 2.  (b) Ablation study on $t_1$ in Eq. 4.

Figure 7: Ablation study on hyper-parameters.

**Ablation on Hyper-parameters.** We utilize content preservation in Eq. 2 to maintain the original content from the source video. By varying the feature injection steps in Fig. 7a, we observe that replacing the value features at a few steps introduces inconsistencies in the background (footprints on the beach). In practice, we adhere to the hyper-parameter selection outlined in [5] (last row). Simultaneously, we note that adjusting the blend layers and steps in Eq. 4 can effectively regulate the extent to which the edited image adheres to the original image. For instance, in the stylization demonstrated in Fig. 7b, injecting the attention map into fewer (15) steps yields a stylized output that may not retain the same structure as the input, while injecting into all 50 steps results in videos with nearly identical textures but less stylization. Users have the flexibility to adjust the blended steps to achieve their preferred balance between stylization and fidelity.

## 6  Conclusion and Limitations

In this paper, we design a novel tuning-free framework UniEdit for both video motion and appearance editing. By leveraging a motion-reference branch and a reconstruction branch and injecting features into the main editing path, it is capable of performing motion editing and various appearance editing. There are nevertheless some limitations. Firstly, we observe performance degradation when performing both types of editing simultaneously. Secondly, since our work is based on T2V models, the proposed method also inherits some of the shortcomings of the existing models, such as inferior performance in understanding complex prompts. We exhibit the failure cases in Appendix B.6.

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

# Supplementary Materials

We organize the Appendix as follows:

- Appendix A: detailed descriptions of experimental settings.
- Appendix B: more experimental results, including:

  - Editing results on different T2V model (Appendix B.1).
  - Quantitative ablation on hyper-parameter selection (Appendix B.2).
  - Ablation study on mask-guided coordination (Appendix B.3).
  - Observation and analysis on the proposed components (Appendix B.4).
  - Analysis and comparison on inference time (Appendix B.5).
  - Failure cases visualization (Appendix B.6).
  - More Comparisons with baseline methods (Appendix B.7).
  - More Editing results of UniEdit (Appendix B.8).

- Appendix C: Broader Impacts.

We encourage the readers to watch the videos on our project page.

## A   Detailed Experimental Settings

**Base T2V Model.**   We instantiate the proposed method on LaVie [63], which is a pre-trained text-to-video generation model that produces consistent and high-quality videos. To achieve a fair comparison, we only leverage the base T2V model in LaVie and load the open-source pre-trained weights for video editing tasks in the experiments. Note that the edited video clip could further be seamlessly fed into the temporal interpolation model and the video super-resolution model to obtain video with a longer duration and higher resolution.

**Video Preprocessing.**   For each input video, we resize it to the resolution of $320 \times 512$, followed by normalization, which is consistent with the training configuration of LaVie. Then, the pre-processed video is fed into the base model of Lavie to perform video editing. To maximize the generation power of LaVie, we set all input videos to 16 frames. For a source video, it takes 1-2 minutes to edit on an NVIDIA A100 GPU.

**Configurations.**   For real source videos, we inverse them with 50 DDIM inversion steps and perform DDIM deterministic sampling with 50 steps for generation. For the generated videos, we use the same start latent of synthesizing the source video as the initial noise $z_T$ for the main editing path and two auxiliary branches. We use the commonly used classifier-free guidance technique [27] with a scale of 7.5.

**Details of User Study.**   As a text-guided editing task, in addition to CLIP scores, it is crucial to evaluate results through human subjective assessment. To achieve this, we utilized MOS (Mean Opinion Score) as our metric and collected feedback from 10 experienced volunteers. We randomly selected 20 editing samples and permuted results from different models. Volunteers were then tasked to evaluate the results based on two perspectives: frame consistency and textual alignment. They provided ratings for these aspects on a scale of 1-5. Specifically, frame consistency measures the smoothness of the video, aiming to avoid dramatic jittering and ensure coherence between the content of each frame. Textual alignment assesses whether the editing results adhere to the text guidance and maintain the content of the source video. In the end, we computed the average user ratings for each method as our final results.

As illustrated in Tab. 1, UniEdit shows the best performance on frame consistency. Regarding textual alignment, UniEdit significantly outperforms all other baselines, demonstrating its capacity to support diverse editing scenarios.

**Baselines.** We implement all baseline methods with their official repositories. For MasaCtrl [5], we adapt it to video editing by first setting the base model to a T2V model [63], then performing MasaCtrl on all frames of the source video. Moreover, since most baselines use StableDiffusion (SD) as the base model, we resize the source video to $512 \times 512$ to align with the default configuration of SD, then feed it into the denoising model, which can maximize the power of SD.

## B Additional Experimental Results and Analysis

### B.1 Results on Different T2V Model

We additionally implement our method on VideoCrafter2 [9], a concurrent work on T2V generation to demonstrate the flexibility of UniEdit. The results are shown in Fig. 8.

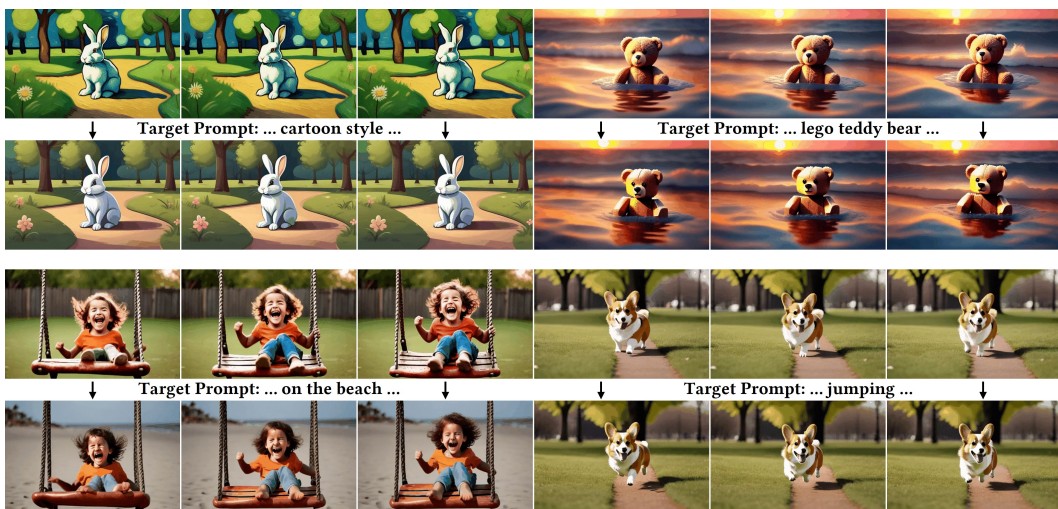

Figure 8: Editing results with UniEdit on VideoCrafter2 [9].

### B.2 Quantitative Ablation on Hyper-parameter Selection

In practice, we empirically found set these values to fixed values, i.e., $t_0 = 50, L = 10$ (same as MasaCtrl [5]) and $t_1 = 25$ can achieve satisfying results on most cases, and we further perform a quantitative study when applying different hyper-parameters in Tab. 3&4.

Table 3: Quantitative comparison on hyper-parameter selection.

| Metric | Frame Similarity | Textual Alignment | Frame Consistency |
|---|---|---|---|
| $t_0 = 20, L = 10$ | 94.33 | 31.57 | 98.09 |
| $t_0 = 50, L = 10$ | 96.29 | 31.84 | 98.12 |
| $t_0 = 50, L = 8$ | 96.76 | 31.25 | 98.11 |

Table 4: Quantitative comparison on hyper-parameter selection.

| Metric | Frame Similarity | Textual Alignment | Frame Consistency |
|---|---|---|---|
| $t_1 = 20$ | 96.21 | 30.92 | 98.06 |
| $t_1 = 25$ | 96.29 | 31.43 | 98.09 |
| $t_1 = 30$ | 96.50 | 31.04 | 98.08 |

## B.3 Ablation Study on the Impact of Mask-Guided Coordination

To investigate the impact of mask-guided coordination, we begin by visualizing masks obtained from 1) the attention map in CA-S modules; 2) the off-the-shelf segmentation model SAM [38], followed by presenting both qualitative and quantitative results of implementing UniEdit with or without mask-guided coordination.

As verified by previous work [24], the attention maps in CA-S modules contain correspondence information between text and visual features. The underlying intuition is that the attention maps between each word and the spatial features at point $(i, j)$ indicate 'how similar this token is to the spatial feature at this location'. We visualize the text-image cross attention map alongside the synthesized frame in Fig. 9. We observe spatial correspondences that align with the video output from the attention map. For instance, areas with higher values of the token 'man' and 'NYC' correspond to the foreground and background, respectively. We further employ a fixed threshold (0.4 in practice) to derive binary segmentation maps from the attention maps. For comparison, we also display the segmentation mask obtained by point prompt on SAM. It's observed that the cross-attention mask is generally accurate and could serve as a reliable proxy in practice when an external segmentor is not available.

We examine the impact of mask-guided coordination through both qualitative and quantitative results across 4 settings: {w/o UniEdit, UniEdit w/o mask, UniEdit with mask from CA-S, UniEdit with mask from SAM}. Qualitatively, shown in Fig. 10, the implementation of UniEdit significantly enhances the consistency between the edited videos and the original video. The application of the mask-guided coordination technique further improves the consistency of unedited areas (e.g., color and texture). The quantitative results in Tab. 5 align coherently with this analysis.

Table 5: Ablation on the proposed mask-guided coordination.

| Metric | Textual Alignment | Frame Consistency |
|---|---|---|
| TAV | 27.89 | 95.39 |
| MasaCtrl* | 25.58 | 97.61 |
| FateZero | 27.30 | 96.72 |
| Rerender | 27.94 | 97.18 |
| TokenFlow | 28.58 | 97.02 |
| UniEdit (w/o mask) | 31.43 | 98.35 |
| UniEdit (w CA-S mask) | 31.49 | 98.33 |
| UniEdit (w SAM mask) | 31.50 | 98.36 |

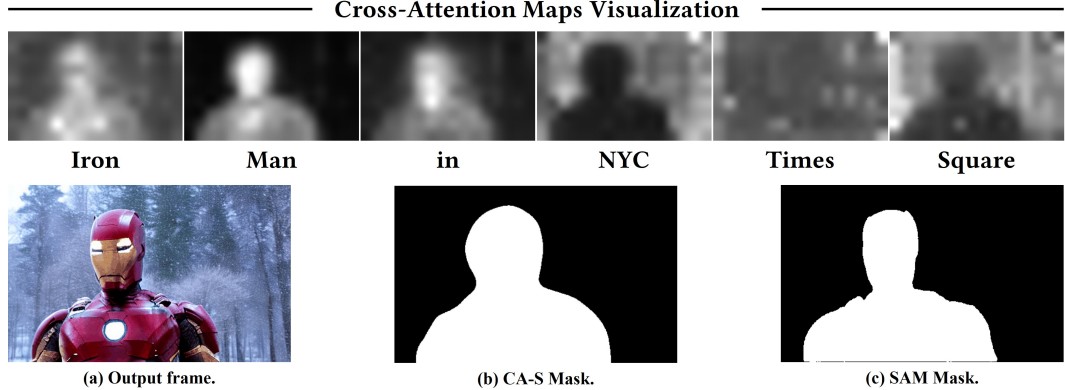

Figure 9: Visualization of attention maps and masks in mask-guided coordination (Sec. 4.3). The top row are attention maps corresponding to different tokens in CA-S modules, (a) is the final output frame, (b) and (c) are the foreground/background binary mask obtained by employing a threshold on the attention map of 'Man' token and point prompt segmentation with SAM, respectively.

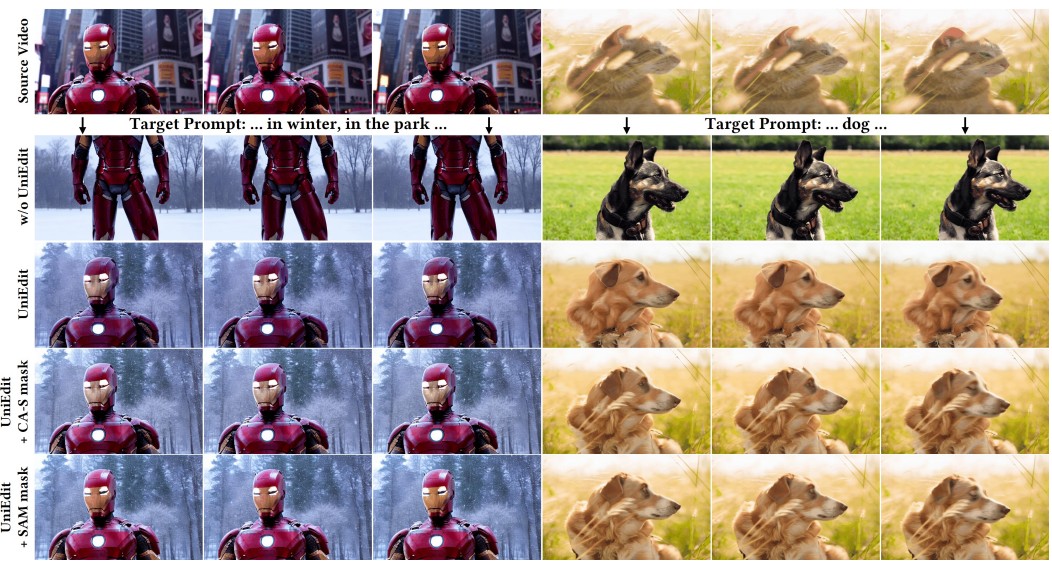

Figure 10: Qualitative editing results across 4 settings: w/o UniEdit (2nd row), UniEdit w/o mask (3rd row), UniEdit with mask from CA-S (4th row), UniEdit with mask from SAM (5th row).

### B.4 More Observation and Analysis on the Proposed Components

**Difference Between QK and V Features in SA-S Modules** To comprehend why we can have inhomogeneous QK and V and their differences, we visualized the results of swapping different features (QK or V) in SA-S modules during style transfer tasks on the source video in Fig. 11a. As can be seen, compared to editing with no feature replacement (2nd row), replacing QK in the 3rd row results in the edited video adopting the same spatial structure as the source video. Simultaneously, replacing V eradicates the style information in the 4th row, meaning the texture details from the source video are utilized to replace the style depicted by the target prompt. To summarize, the query and key features (in SA-S modules) dictate the spatial structure of the generated video, while the value features tend to influence the texture, including details such as color tones.

**Influence of Spatial Structure Control in Motion Editing** We explored the role of spatial control in motion editing. The proposed method synthesizes videos with larger modifications when removing the spatial control mechanism on both the motion-reference branch and the main editing branch. We visualized the results in Fig. 11b. It can be observed that although the motion-reference branch can still generate the target motion without the control of spatial structure, the layout deviates significantly, for example, the raccoon assumes a different pose and location. We regard this as a suboptimal solution because, compared to the results presented in the 3rd row, the results w/o spatial structure control modifies the object position of the source video, leading to a decrease in consistency between the edited result and the source video.

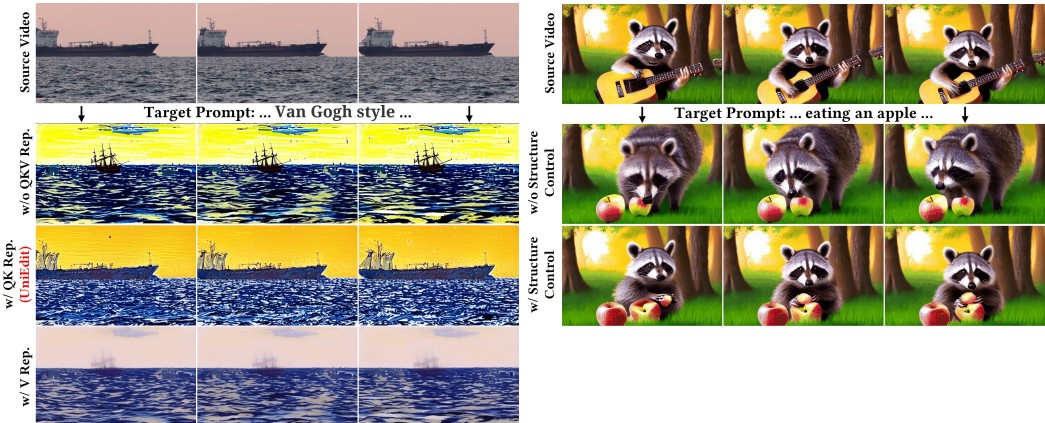

(a) Replacing different features in SA-S modules.  (b) Motion editing w/ or w/o structure control.

Figure 11: Ablation on the proposed feature injection techniques. (11a): comparison of appearance editing without feature replacement (2nd row), with QK replacement (3rd row), with V replacement (4nd row); (11b): comparison of motion editing with and without the designed spatial structure control mechanism.

### B.5 Analysis and Comparison on Inference Time

We conduct a theoretical analysis of the additional cost of UniEdit and an empirical comparison with baseline methods in terms of inference speed.

Theoretically, our method primarily involves feature replacement operations in attention modules, achieved through forward hook registration and introducing minimal additional computation. Therefore, the main difference between synthesizing a video from random noise and editing a video with UniEdit lies in the batch size of the denoising process (i.e., vanilla generation: batchsize=1, appearance editing: batchsize=2, motion editing: batchsize=3), and this process could be further accelerated through multi-GPU parallel processing techniques. Additionally, we utilize LaVie [63] as the base T2V model in the paper, which takes approximately 45 seconds to synthesize a 16-frame video. Our method can be even faster when adapted to more efficient base models.

Empirically, UniEdit demonstrates comparable speed with baseline methods. The comparison of inference time on a single 16-frame source video clip with a resolution of 320x512 on 1 NVIDIA A100 GPU is as follows:

Table 6: Quantitative comparison on inference time of editing a single 16-frame video clip.

| Method | TAV | MasaCtrl* | FateZero | Rerender | TokenFlow | UniEdit (appearance editing) | UniEdit (motion editing) |
|---|---|---|---|---|---|---|---|
| Inference time | ∼10min | ∼90s | ∼130s | ∼110s | ∼100s | ∼95s | ∼125s |

### B.6 Failure Cases Visualization

We exhibit failure cases in Fig. 12. Fig. 12a showcase when editing multiple elements simultaneously, and we observe a relatively large inconsistency with the source video. A naive solution is to perform editing with UniEdit multiple times. Fig. 12b visualizes the results when editing video with complex scenes, and the model sometimes could not understand the semantics in the target prompt, resulting in incorrect editing. This may be caused by the base model's limited text understanding power, as discussed in [30]. It could be alleviated by leveraging the reasoning power of MLLM [30], or adapting approaches in complex scenario editing [40].

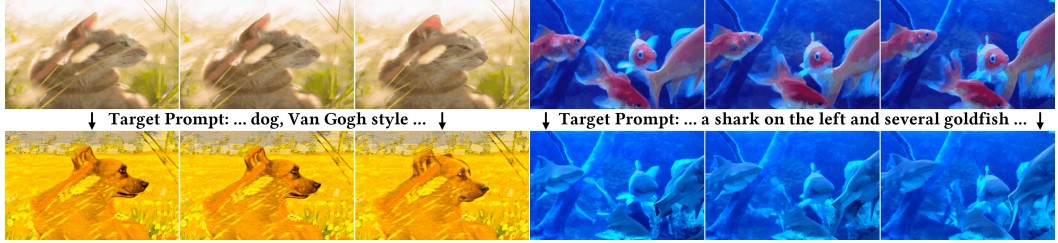

(a) Edit multiple elements simultaneously.    (b) Complex scene editing.

Figure 12: Visualization of failure cases.

### B.7 More Comparison with State-of-the-Art Methods

Please refer to Fig. 13 and Fig. 14 for more comparison with the state-of-the-art methods. For a fair comparison, we also migrated all baselines to LaVie [63], using the same base model as our method. The results are presented in Fig. 15, and they are found to be inferior compared to those in Fig. 5 (based on Stable Diffusion).

### B.8 More Results of UniEdit

More edited results of UniEdit are provided in Fig. 16-21. Examples of TI2V generation are provided in Fig. 22.

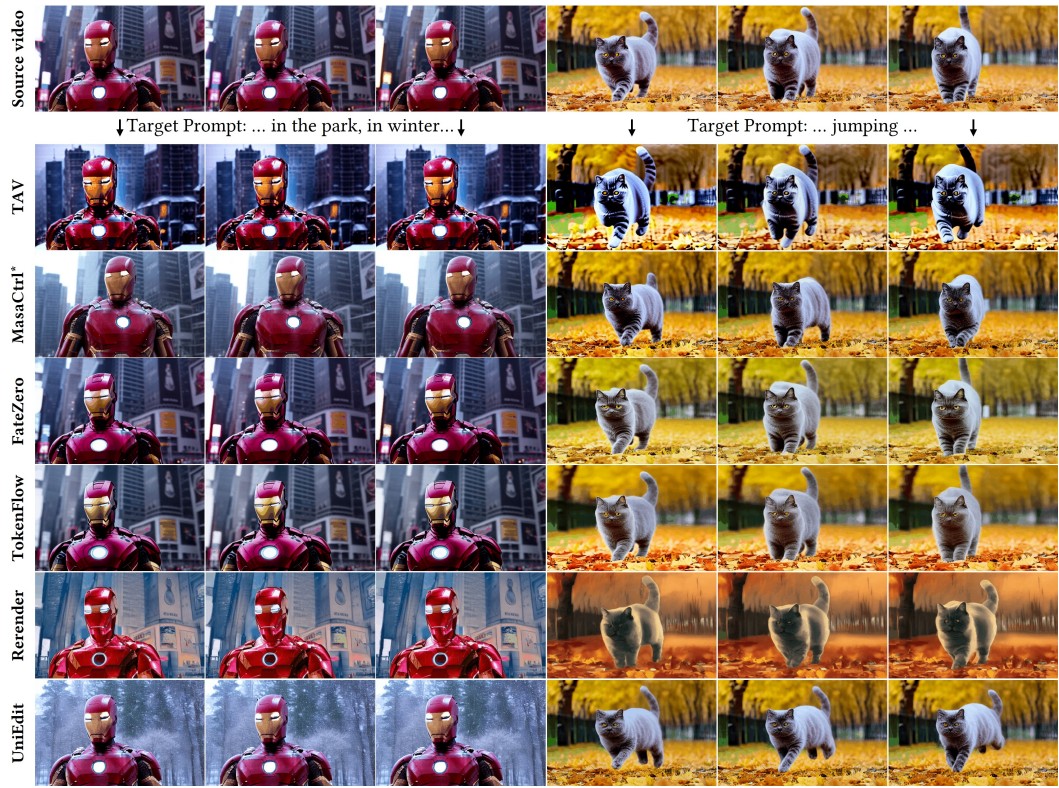

Figure 13: More comparison with state-of-the-art methods.

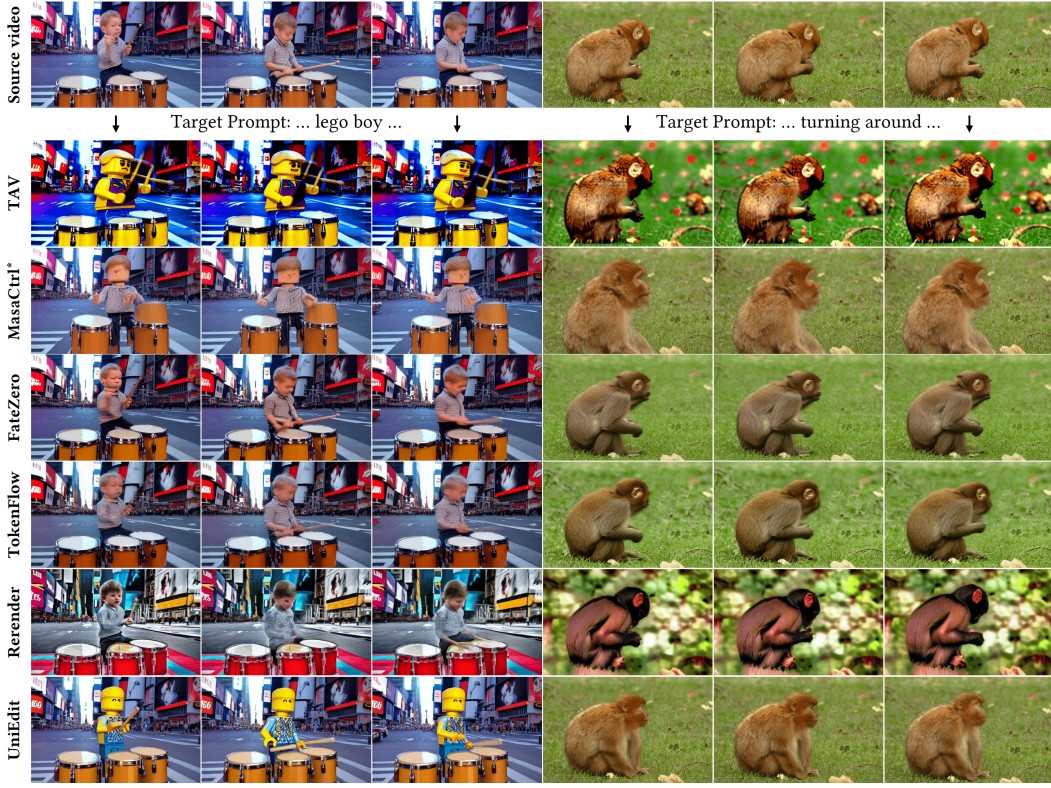

Figure 14: More comparison with state-of-the-art methods.

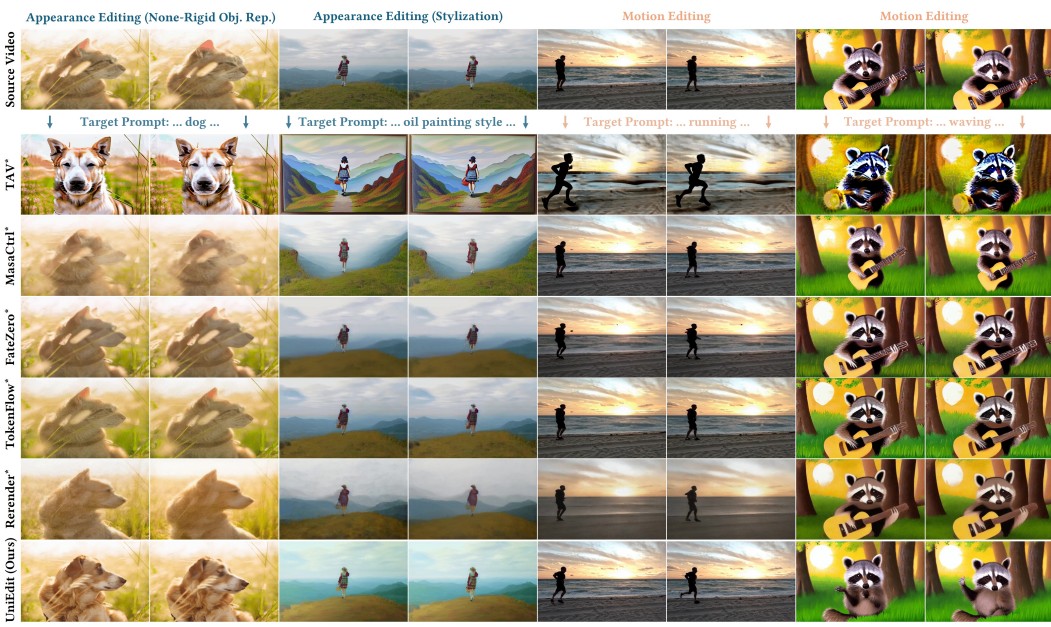

Figure 15: More comparison with state-of-the-art methods. We adapt the baseline methods to the text-to-video model LaVie [63] and compare with our method (also based on LaVie).

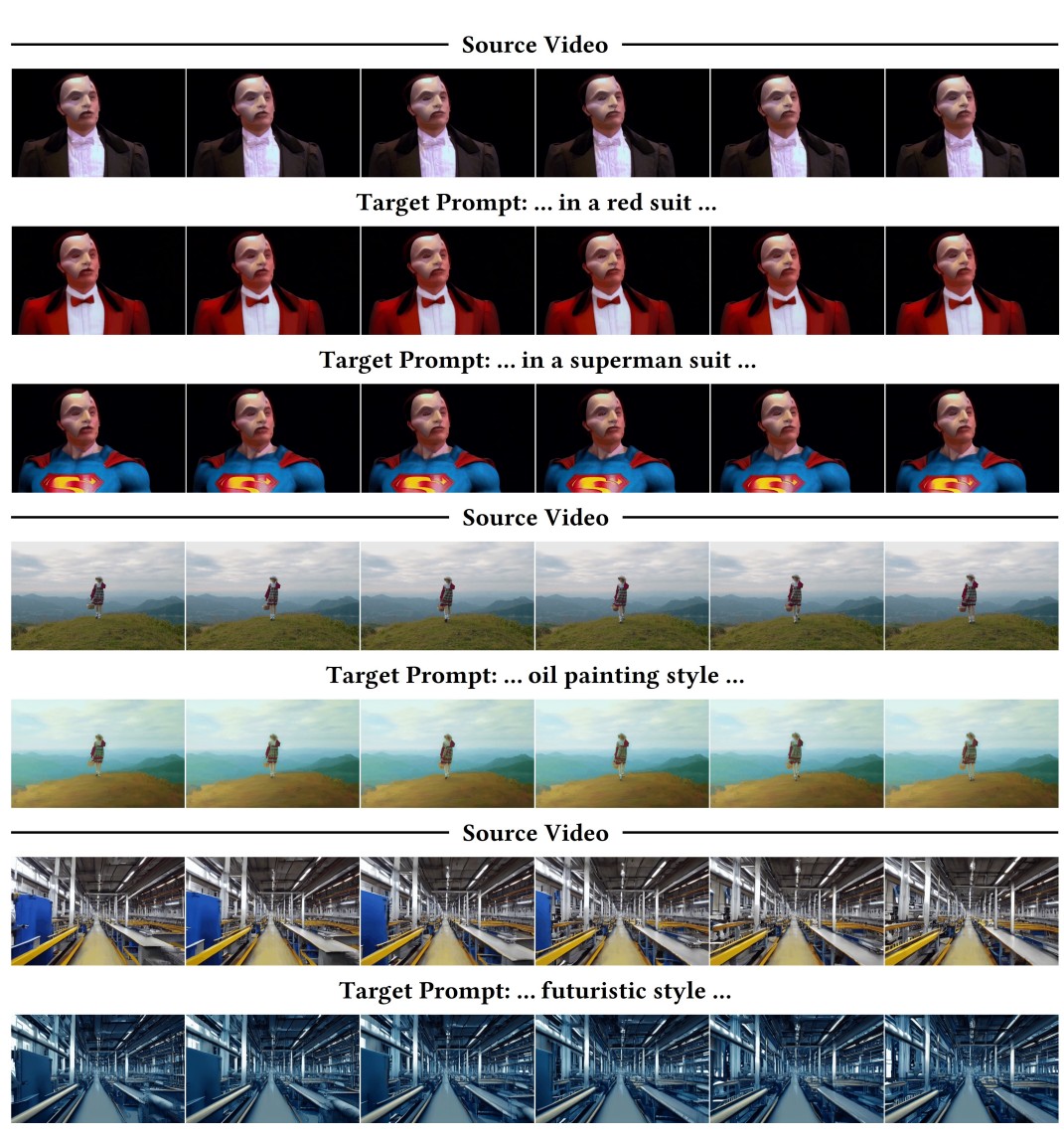

Figure 16: More appearance editing results of UniEdit.

**Source Video**

**Target Prompt: ... black and white ...**

**Target Prompt: ... at night ...**

**Source Video**

**Target Prompt: ... metal robotic ...**

**Target Prompt: ... cute panda ...**

**Target Prompt: ... cute Iron Man ...**

**Target Prompt: ... cute Spider Man ...**

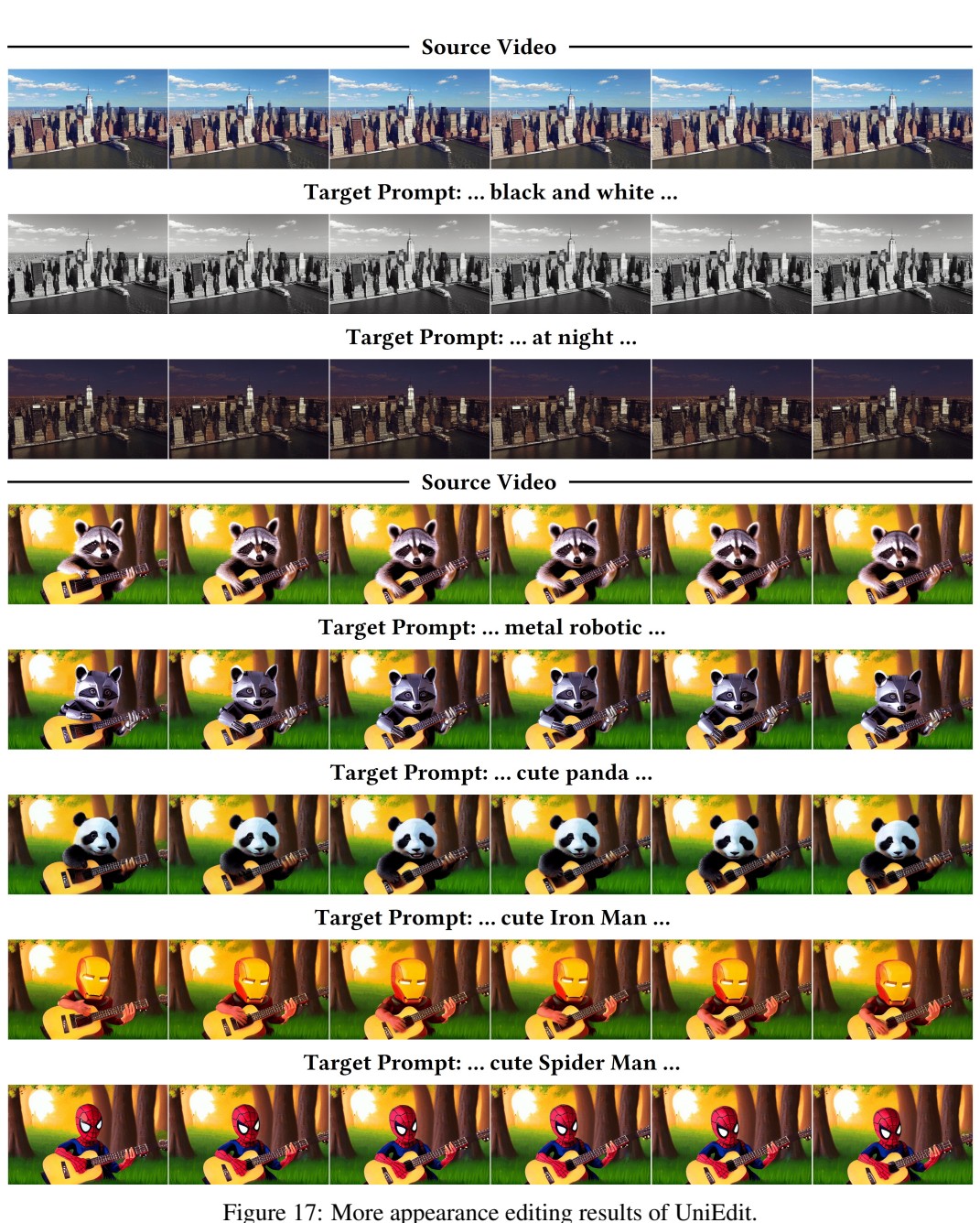

Figure 17: More appearance editing results of UniEdit.

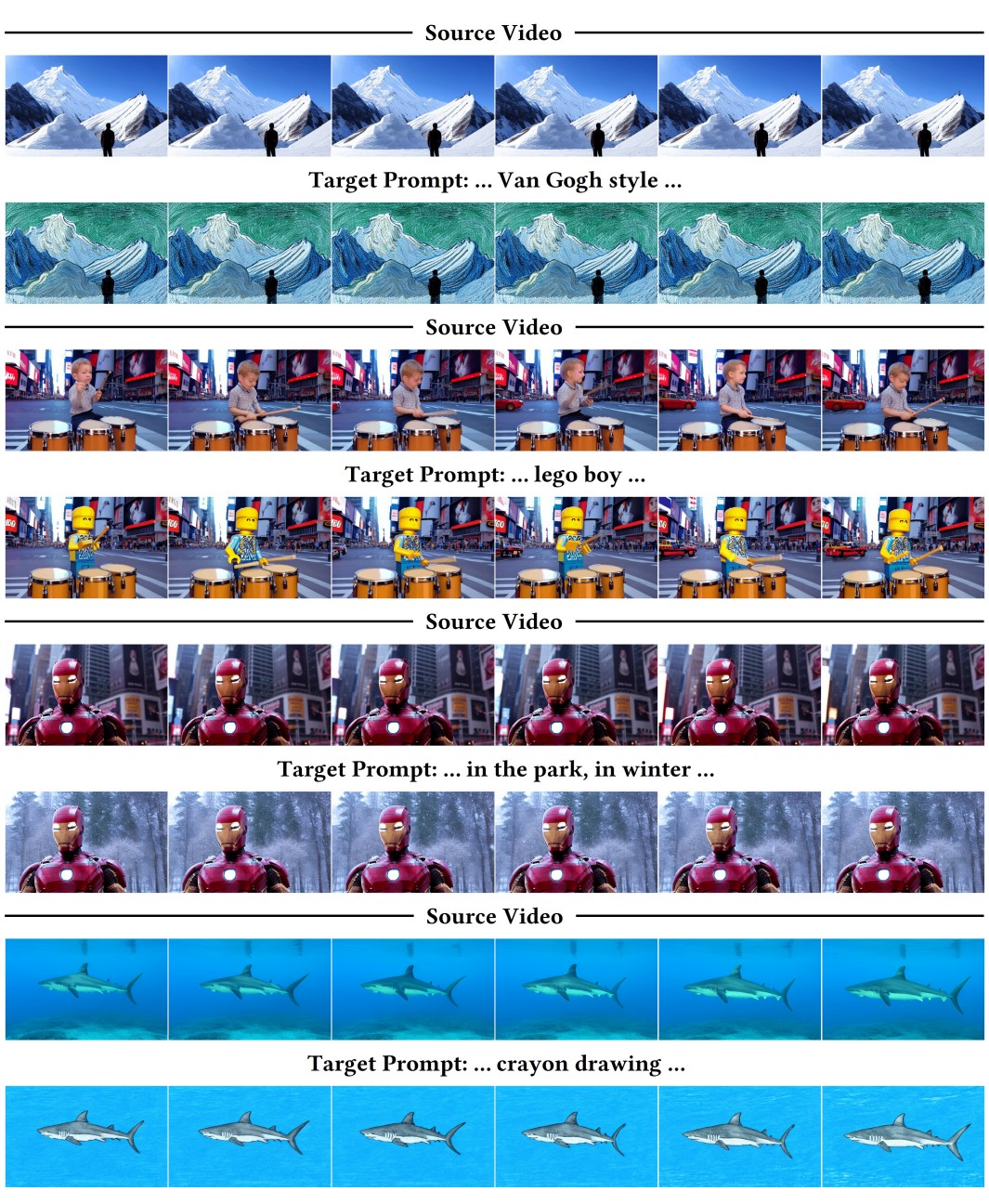

Figure 18: More appearance editing results of UniEdit.

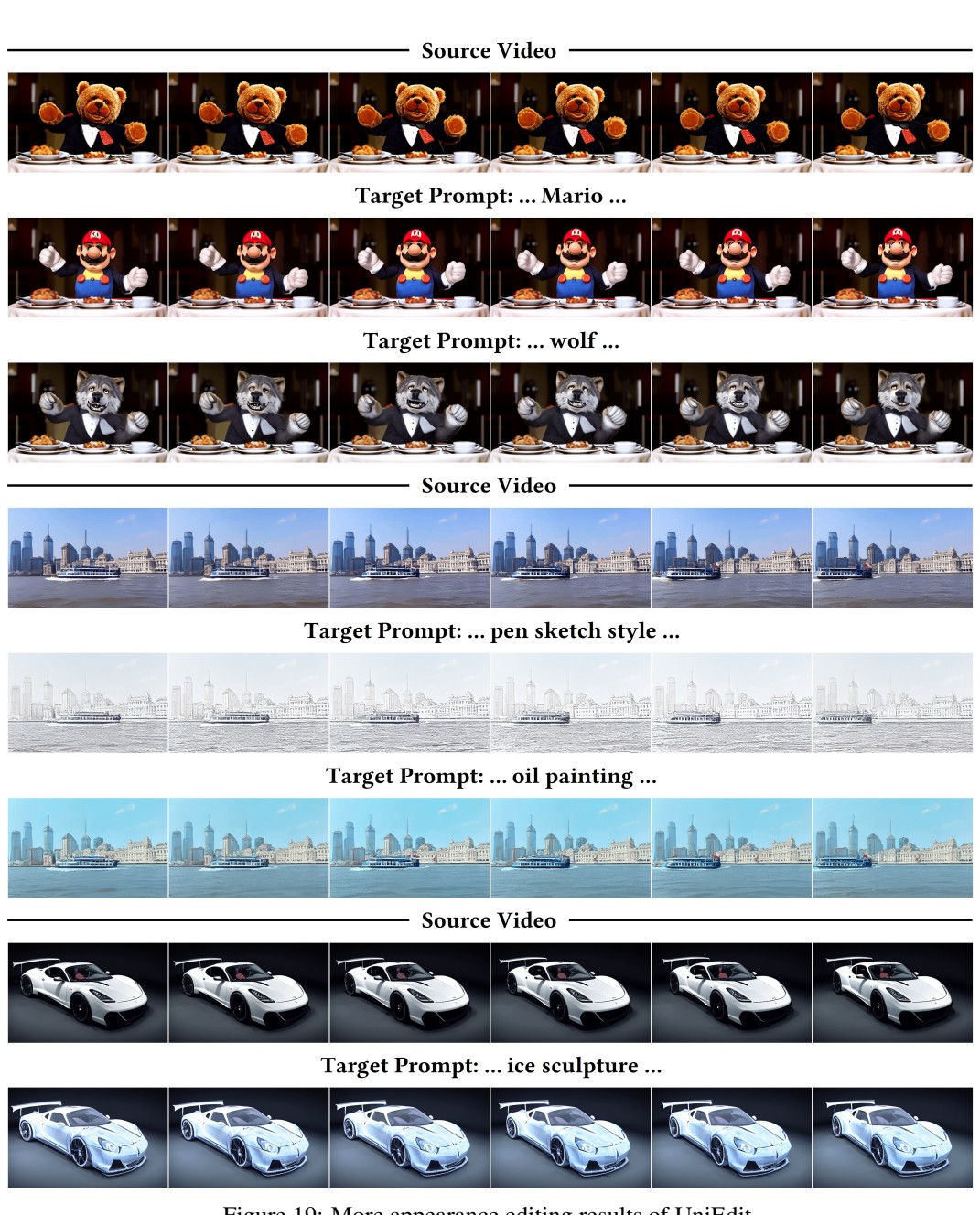

Figure 19: More appearance editing results of UniEdit.

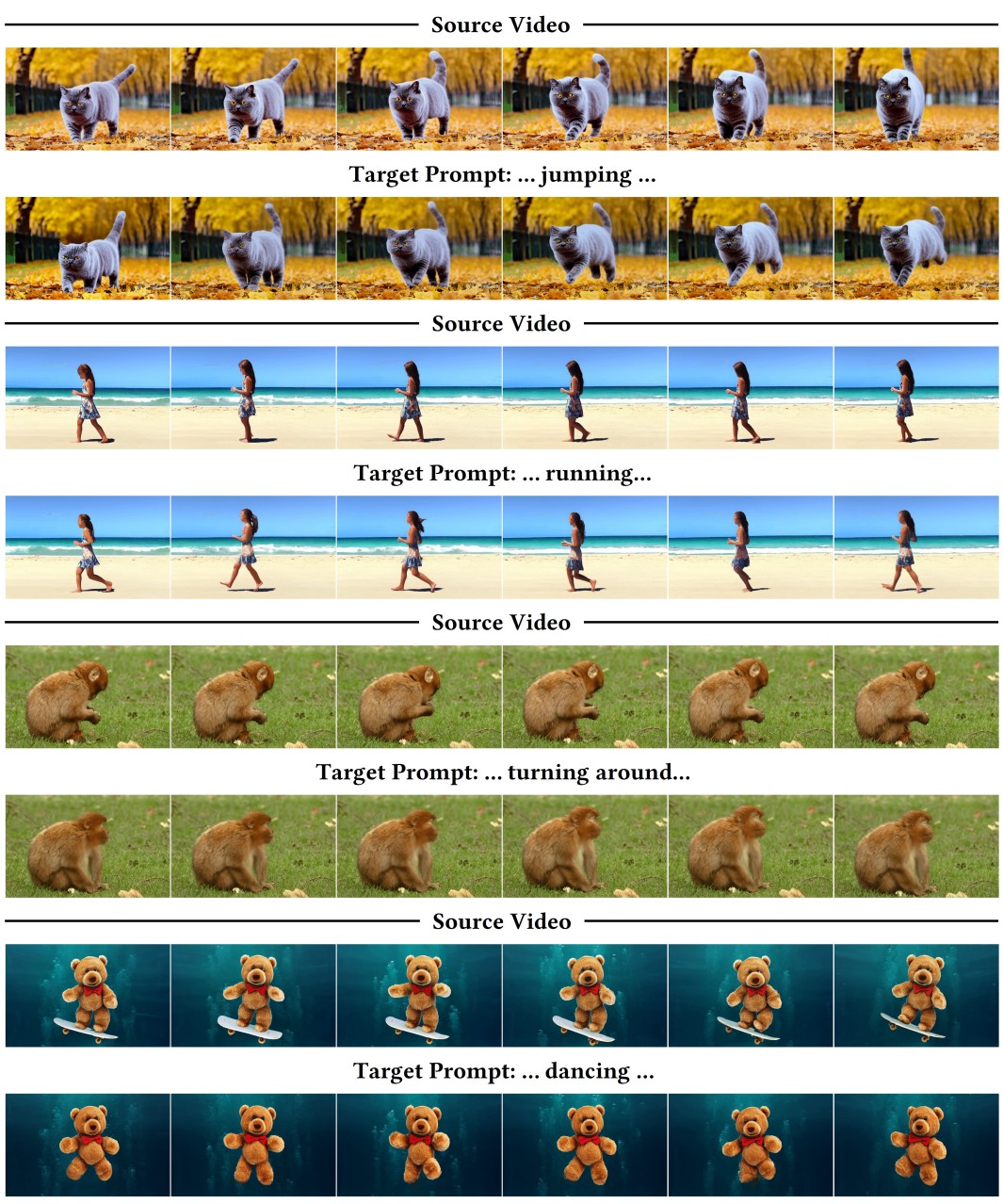

Figure 20: More motion editing results of UniEdit.

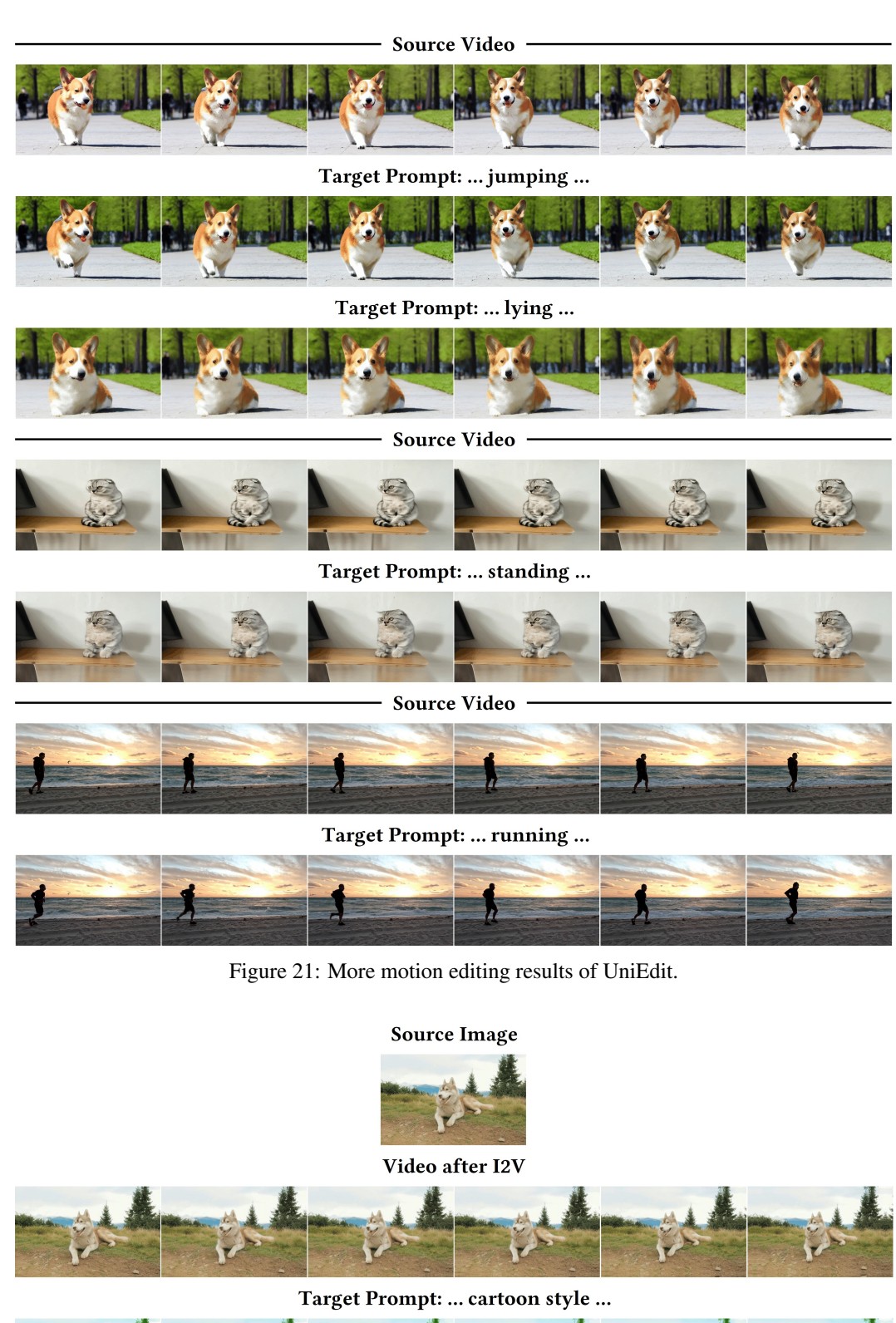

Figure 21: More motion editing results of UniEdit.

**Source Image**

**Video after I2V**

**Target Prompt: ... cartoon style ...**

Figure 22: Results of text-image-to-video synthesis in Sec. 4.4.

## C  Broader Impacts

UniEdit is a tuning-free approach and is intended for advancing AI/ML research on video editing. We encourage users to use the model responsibly. We discourage users from using the codes to generate intentionally deceptive or untrue content or for inauthentic activities. It is suggested to add watermarks to prevent misuse.

