# OpenReview forum: "UniEdit: A Unified Tuning-Free Framework for Video Motion and Appearance Editing"
_NeurIPS.cc/2024/Conference — Submitted to NeurIPS 2024_

### Official Review · Reviewer_t6q7 · 2024-06-20

**Soundness:** 3
**Presentation:** 3
**Contribution:** 4
**Rating:** 5
**Confidence:** 1

**Summary:**

The paper proposes UniEdit, a framework that allows the editing of videos. More specifically, UniEdit allows manipulation via text prompts to change the visual style or the motion pattern that is visible in the video. Moreover, it also targeted steering, e.g. via segmentation masks. They achieve this by introducing an additional reconstruction branch and a motion-reference branch into a u-net based diffusion network and share the values of the attention layers which are party designed specifically for this work. The method allows editing videos without retraining and creates very good results.

**Strengths:**

The core idea is straightforward and well-presented. There are just a few hyperparameters to select. They have a large amount of visual content showing the quality of their contribution. Moreover, they performed extensive human experiments to rate the videos.

**Weaknesses:**

The implementation may not be entirely reconstructable. Hopefully, this issue will be fixed when they publish the code as promised.

Even if the method description is understandable, the math is sometimes not entirely correct. For example in line 212/212, M is a matrix but the notation says that it is a scalar from a set $\{ -\inf, 1 \} $ or $\{ 0, 1 \} $. The authors should be encouraged to revise the math present in the paper.

**Minor weaknesses:**
Sometimes the English writing is a bit weak. For example:
 - 100-110: Some parts are not forming complete sentences, e.g. "Other improvements like efficiency [1], training strategy [19], or additional control signals [16], etc."
 - 187: I think it should be "an additional network" or "additional networks"

**Questions:**

- Is there a reference that also handles the evaluation process as described in lines 242-245?

**Limitations:**

Unfortunately, there is no benchmark to evaluate the method. This is not the author's fault and they tried to do their best to create baselines. However, this makes it harder to rate the results.

---

> ### Author Rebuttal · Authors · 2024-08-06
>
> We thank the reviewer for providing encouraging comments on our paper! We provide clarifications to the concerns below:
> &nbsp;
>
> > The implementation may not be entirely reconstructable. Hopefully, this issue will be fixed when they publish the code as promised.
>
> We will definitely release the code when it is published. We also make detailed descriptions in Section 4 and Appendix A on experimental settings (e.g., model structure, hyper-parameter selection) to make the paper reconstructable.
>
> &nbsp;
>
> > The math is sometimes not entirely correct. For example in line 212/212, M is a matrix but the notation says that it is a scalar from a set −inf,1 or 0,1. The authors should be encouraged to revise the math present in the paper.
>
> Thanks for pointing it out! We will revise to $ M_{ij} \in \\{ ... \\} $ and check the math expressions in the paper comprehensively.
>
> &nbsp;
>
> > Minor weaknesses in writing.
>
> We would like to thank the reviewer for a very detailed review! We will fix the typos.
>
> &nbsp;
>
> > Is there a reference that also handles the evaluation process as described in lines 242-245?
>
> Yes. CAMEL [1] uses a subset of LOVEU-TGVE-2023 for evaluating the designed training-based video editing technique, and Customize-A-Video [2] uses the LOVEU-TGVE-2023 dataset for evaluating the motion customization ability.
>
> &nbsp;
>
> > limitation on rating the proposed method.
>
> As you mentioned, the lack of a commonly used benchmark complicates the comparison of the proposed method with existing baselines. In light of this, we evaluate the proposed method in three aspects:
>
> 1. **Following previous work [3]**, we present the CLIP scores and user preferences for UniEdit alongside baseline methods (Tables 1 & 2).
> 2. **We additionally calculate VBench [4] scores**, a recently proposed benchmark suite for T2V models, for a more comprehensive and precise quantitative comparison (Table 1).
> 3. **We provide plenty of qualitative results** in the paper and synthesized videos on our project website to aid readers in making subjective evaluations.
>
> We believe that these are sufficient to demonstrate the superiority of our method.
>
> &nbsp;
>
> We hope the above responses address your concern. **We would be grateful if you would kindly let us know of any other concerns and if we could further assist in clarifying any other issues.**
>
> &nbsp;
>
> [1] Zhang, Guiwei, et al. "CAMEL: CAusal Motion Enhancement tailored for Lifting Text-driven Video Editing." *Proceedings of the IEEE/CVF Conference on Computer Vision and Pattern Recognition*. 2024.
>
> [2] Ren, Yixuan, et al. "Customize-a-video: One-shot motion customization of text-to-video diffusion models." *arXiv preprint arXiv:2402.14780* (2024).
>
> [3] Wu, Jay Zhangjie, et al. "Tune-a-video: One-shot tuning of image diffusion models for text-to-video generation." *Proceedings of the IEEE/CVF International Conference on Computer Vision*. 2023.
>
> [4] Huang, Ziqi, et al. "Vbench: Comprehensive benchmark suite for video generative models." *Proceedings of the IEEE/CVF Conference on Computer Vision and Pattern Recognition*. 2024.

---

### Official Review · Reviewer_UJWt · 2024-07-12

**Soundness:** 3
**Presentation:** 3
**Contribution:** 2
**Rating:** 5
**Confidence:** 4

**Summary:**

This paper suggests UniEdit, a tuning-free method for editing the motion of a given video. The authors use a pre-trained text-to-video diffusion model and utilize its motion prior, to performing motion editing on a video while keeping the appearance of the original video. During the denoising process, they apply structural/content features injection from the reconstruction branch of the original video to maintain the input video's structure or content. The motion is edited according to a text description used to denoise another reference branch which is then used for injecting features into the editing videos. The results improve over the existing methods.

**Strengths:**

* Successfully applying feature injection for video diffusion models.
* Impressive results.

**Weaknesses:**

1. Novelty.  Feature injection for image editing is a known technique [56].  Applying injection to video models is important and challenging, but not novel enough in my view.
Showing that the injection of motion features from the reference motion branch into the edited video, constrains the output motion, is important, but not surprising given the observation of [56].

2. Given that the main insight of the paper is that “the temporal self-attention layers of the generator encode the inter-frame dependency”, there is not enough analysis of this besides the visual results and Figure 6, which shows the relation between the optical flow magnitude and the temporal attention on one example qualitatively. Showing a quantitative analysis, and analyzing the features during the denoising process for different layers, could support this claim better and show the importance of this insight.

**Questions:**

Can the authors respond to the two weaknesses written above?

**Limitations:**

Yes, the authors discuss both, limitations, and broader impact.

---

> ### Author Rebuttal · Authors · 2024-08-06
>
> Thanks for your constructive comments! We address the concerns below:
> &nbsp;
>
> > Feature injection is a known technique in image editing.
>
> As you mentioned, similar feature injection techniques have been explored previously. However, our approach differs in several key ways:
>
> 1. We address the novel challenging problem of motion editing in the temporal dimension, which cannot be achieved by simply adapting existing feature injection methods.  We adapt the SOTA non-rigid image-editing technique MasaCtrl [1] for video editing, and it fails to synthesize video that adheres to the text prompt as illustrated in Fig. 5 and Tab. 1.
>
> 2. We provide insight that “the temporal attention layers of the generator encode the inter-frame dependency”, enabling training-free motion editing. Building upon this, we explore and investigate feature injection in temporal layers, an area that has not been thoroughly explored.
>
> 3. Furthermore, simply performing feature injection on temporal layers results in severe content inconsistency with the source video (Tab. B). In response, we design UniEdit with content preservation and structure control on spatial layers and motion injection on temporal layers.
>
> 4. Previous works in video editing are typically tailored to particular tasks. For instance, Rerender-A-Video [2] excelled in style transfer, while Video-P2P [3] focused on local object editing. In contrast, our proposed method can effectively handle motion editing and various appearance editing tasks, showcasing remarkable performance both visually (https://uni-edit.github.io/UniEdit/) and quantitatively (Tab. 1) across these domains.
>
> Thus we believe that UniEdit contributes to the advancement of video editing.
>
> &nbsp;
>
> > Showing a quantitative analysis, and analyzing the features, could support this claim better and show the importance of the insight “the temporal self-attention layers of the generator encode the inter-frame dependency”.
>
> Thanks for the advice! We add additional analysis and results to further support our core insight as follows:
>
> 1. **Quantitative results.** We calculate the difference between the attention map of temporal layers and the optical flow. The intuition behind this is that optical flow captures the movement between two consecutive frames at the pixel level, and we assume temporal layers capture inter-frame dependency in the feature space. Hence, we sample 40 videos and their corresponding 640 frames, and report the average $L_1$ distance and KL divergence:
>
>    | Distance Metrics   | $L_1$ Distance | KL Divergence |
>    | ------------------ | :------------: | :-----------: |
>    | Random Matrix      |      0.51      |     1.33      |
>    | Spatial-Attention  |      0.48      |     1.14      |
>    | Temporal-Attention |      0.29      |     0.81      |
>
>    It's observed that attention maps from the temporal layers are more similar to the magnitude of optical flow (also visualized in Fig. A in the rebuttal pdf), which supports the hypothesis quantitively.
>
> 2. **Analyzing features of different layers.** Furthermore, we visualize 1) temporal-attention maps at different resolutions and denoising steps and 2) the feature of {spatial/cross/temporal} attention layers in Fig. A and Fig. B respectively.
>
>    In Fig. A, we visualize temporal attention maps between frame $i$ and frame $i+1$ at different layers (resolution) and denoising steps. In the example on the left, it's observed that the attention map values are higher over the walking person and the moving waves, while they are lower over the relatively static sky and beach. The high consistency with the optical flow across layers and timesteps is consistent with the quantitative results and indicates our insights: the temporal attention layers encode the inter-frame dependency.
>
>    In Fig. B, we contrast the characteristics of SA-S, CA-S, and SA-T. SA-S captures the overall spatial structure of the generated frame, CA-S is mainly activated on the area according to the text, while SA-T concentrates on the inter-frame variances. Leveraging these insights, we delicately design UniEdit to achieve content preservation and structure control by feature injection on SA-S layers and motion transferring on SA-T layers.
>
> 3. **Synthesized frames visualization when scaling the temporal features.** To demonstrate temporal layers modeling inter-frame dependency, we multiply the output of the temporal layer in each block by a constant factor $\gamma$ before adding it back to the input features $x_{in}$, i.e., $x_{out}=x_{in} + \gamma * \texttt{SA-T}(x_{in})$. The synthesized frames are exhibited in Fig. C. When $\gamma$ is 0, there is no interaction between frames, resulting in no correlation among the generated video frames. The inter-frame consistency strengthens as $\gamma$ increases, which confirms our hypothesis.
>
> **Note: The corresponding videos of Fig. A, B, and C are available at https://uni-edit.github.io/UniEdit/#sectionRebuttal**
>
> We hope the above responses address your concern. **We would be grateful if you would kindly let us know of any other concerns and if we could further assist in clarifying any other issues.**
>
> &nbsp;
>
> [1] Cao et al. "Masactrl: Tuning-free mutual self-attention control for consistent image synthesis and editing." ICCV'23.
>
> [2] Yang et al. "Rerender a video: Zero-shot text-guided video-to-video translation." SIGGRAPH Asia'23.
>
> [3] Liu et al. "Video-p2p: Video editing with cross-attention control." CVPR'24.

---

> > ### Comment · Reviewer_UJWt · 2024-08-11
> >
> > In my opinion, Content Preservation, Motion Injection, and Spatial Structure Control (Equations 2, 3, and 4) are all different forms of feature injection and a direct extension of feature injection [56] from image diffusion models to video diffusion models. While I appreciate the quality of the results and the technical contribution, I do not find it particularly novel. However, I find the analysis of the temporal self-attention layers provided here interesting. Assuming this analysis will be included in the revised version, I would like to increase my score to borderline accept.

---

> > > ### Author Response · Authors · 2024-08-12
> > > **Further Response to Reviewer UJWt**
> > >
> > > Thank you for your response and appreciation! We are dedicated to continually enhancing our work, and we will incorporate these analyses and experiments into the final version.

---

### Official Review · Reviewer_rXMa · 2024-07-15

**Soundness:** 2
**Presentation:** 3
**Contribution:** 3
**Rating:** 6
**Confidence:** 2

**Summary:**

This paper focuses on developing a tuning-free framework capable of editing both the motion and appearance of videos. They introduce UniEdit, an approach designed for text-guided motion editing that maintains the original content of the source video. By utilizing two branches—an auxiliary reconstruction branch and an auxiliary motion-reference branch—they achieve both content preservation and effective motion editing.

**Strengths:**

1. This paper pointing out the problem of existing methods of not being able to keep the non-edited area and propose using spatial self-attention module, spatial cross-attention module and temporal self-attention model to solve the problem. From the experiment results, it shows that the edited results exhibit the editing task correctly while maintaining the unedited area.
2. The paper is overall clear and well-written
3. This paper provides versatile applications like motion editing, stylization, rigid/non-rigid object editing, and background editing.

**Weaknesses:**

1. The number of the participants in the user study might not be representative enough.

**Questions:**

1. The first row of the table 2 is missing. What is that?
2. It will be more informative if you could ablate with motion injection and structure control solely in Table 2 as well.
3. How do you measure texture alignment and frame consistency in Table 2?

**Limitations:**

This method is inherently influenced by the T2V model used.

---

> ### Author Rebuttal · Authors · 2024-08-06
>
> Thanks for the elaborate review! We will address your concerns below:
> &nbsp;
>
> > The number of the participants in the user study might not be representative enough.
>
> Thanks for the advice! We additionally recruited 20 participants to make their evaluations on the synthesized videos of the proposed UniEdit and baselines, the average rating scores of 30 participants (same as Rerender-A-Video [1]) are as follows in Tab. A.  It's observed that UniEdit outperforms baselines on temporal consistency and alignment with the target prompt. We will update Tab. 1 in the final version.
>
> Tab. A: User preference comparison with state-of-the-art video editing techniques.
>
> | Method       | Frame Consistency | Textual Alignment |
> | ------------ | :---------------: | :---------------: |
> | TAV          |       3.75        |       3.32        |
> | MasaCtrl*    |       4.34        |       3.13        |
> | FateZero     |       4.49        |       3.49        |
> | Rerender     |       4.16        |       3.57        |
> | TokenFlow    |       4.51        |       3.30        |
> | **UniEdit**      |       **4.73**        |       **4.77**        |
> | **UniEdit-Mask** |       **4.76**        |       **4.91**        |
>
> &nbsp;
>
> > The first row of the table 2 is missing. What is that?
>
> The first row refers to not using any of the three components, i.e., performing vanilla text-to-video generation conditioned on the target prompt. As shown in Fig. 6b, vanilla text-to-video generation results in content inconsistencies when compared to the source video. The proposed modules notably enhance the editing results on both quantitative results in Tab. 1&2 and qualitative results on the project website. We will modify Tab. 2 with Tab. B to make it informative and easy to understand  in the final version.
>
> &nbsp;
>
> > It will be more informative if you could ablate with motion injection and structure control solely in Table 2 as well.
>
> Thanks for the advice! We complement the results below. It's observed that the use of motion injection significantly enhances ‘Textual Alignment’, indicating successful transfer of the targeted motion to the main editing path, and structure control mainly contributes to the ‘Frame Similarity’ metric. The best results are achieved through the combined use of all components.
>
> Tab. B:  Impact of various components.
>
> | Content Preservation | Motion Injection | Structure Control | Frame Similarity | Textual Alignment | Frame Consistency |
> | -------------------- | :--------------: | :---------------: | ---------------- | ----------------- | ----------------- |
> | -                    |        -         |         -         | 90.54            | 28.76             | 96.99             |
> | &#x2713;             |        -         |         -         | 97.28            | 29.95             | 98.12             |
> | -                    |     &#x2713;     |         -         | 90.66            | 31.49             | 98.13             |
> | -                    |        -         |     &#x2713;      | 90.68            | 29.99             | 98.10             |
> | -                    |     &#x2713;     |     &#x2713;      | 91.30            | 31.48             | 98.08             |
> | &#x2713;             |     &#x2713;     |         -         | 96.11            | 31.37             | 98.12             |
> | &#x2713;             |     &#x2713;     |     &#x2713;      | 96.29            | 31.43             | 98.09             |
>
> &nbsp;
>
> > How do you measure texture alignment and frame consistency in Table 2?
>
> We will add the explanation below in the final version:
>
> To evaluate the adherence of the edited video $V$ ($N$ frames) to the target prompt $P_t$, we follow previous work [2] to compute the average similarity of the CLIP embedding of the edited video clip and its corresponding text prompt, a metric we refer to as 'Textual Alignment'. To quantify frame consistency, we calculate the avarage cosine similarity of CLIP embeddings between adjacent frames. Formally, the two metrics are defined as:
>
> $ \text{Textual Alignment} = \frac{1}{N} \sum_{i=1}^{N} \frac{\text{CLIP}(V_i) \cdot \text{CLIP}(P_t)}{\|\text{CLIP}(V_i)\| \cdot \|\text{CLIP}(P_t)\|} $
>
> $ \text{Frame Consistency} = \frac{1}{N-1} \sum_{i=1}^{N-1} \frac{\text{CLIP}(V_i) \cdot \text{CLIP}(V_{i+1})}{\|\text{CLIP}(V_i)\| \cdot \|\text{CLIP}(V_{i+1})\|} $
>
> &nbsp;
>
> We hope the above responses address your concern. **We would be grateful if you would kindly let us know of any other concerns and if we could further assist in clarifying any other issues.**
>
> &nbsp;
>
> [1] Yang, Shuai, et al. "Rerender a video: Zero-shot text-guided video-to-video translation." *SIGGRAPH Asia 2023 Conference Papers*. 2023.
>
> [2] Wu, Jay Zhangjie, et al. "Tune-a-video: One-shot tuning of image diffusion models for text-to-video generation." *Proceedings of the IEEE/CVF International Conference on Computer Vision*. 2023.

---

> > ### Comment · Reviewer_rXMa · 2024-08-12
> >
> > Thanks for the efforts of addressing all the questions and concerns. I have no further questions, and will keep the current rating.

---

> > > ### Author Response · Authors · 2024-08-12
> > > **Further Response to Reviewer rXMa**
> > >
> > > Thanks for your reply! We are delighted to see that our responses addressed your questions and concerns. We will incorporate these classifications and experiments into the final version.

---

### Author Rebuttal · Authors · 2024-08-06

# General Response to All Reviewers

&nbsp;

Dear Reviewers:

&nbsp;

**We would like to thank you for the constructive comments and the time you dedicate to the paper!**

We are encouraged to see that UniEdit is acknowledged to address an important problem (Reviewer rXMa) and presents an effective approach to video editing tasks (Reviewer rXMa, Reviewer UJWt), with strong qualitative results (Reviewer rXMa, Reviewer UJWt, Reviewer t6q7). The paper is also well-written (Reviewer rXMa, Reviewer t6q7) and includes high-quality illustrations (Reviewer t6q7).

We have comprehensively supplemented additional experiments and analysis based on your comments, and we hope this addresses your concerns. The figures A, B, and C mentioned in the rebuttal have been included in the **rebuttal PDF document**, and we have also placed the video results on the anonymous project page, **Section: Rebuttal** **(https://uni-edit.github.io/UniEdit/#sectionRebuttal)** for your reference. We would be grateful if you would kindly let us know of any other concerns and if we could further assist in clarifying any other issues.

&nbsp;

Thanks a lot again, and with best wishes

Authors

---

### Decision · Program_Chairs · 2024-09-25

**Decision:**

Reject

**Comment:**

The paper initially received mixed reviews, with ratings of 4, 6, and 7.
**Reviewer UJWt** increased the score to *borderline accept* after the rebuttal but expressed some concerns about the novelty. Specifically, **Reviewer UJWt** noted that *"Content Preservation, Motion Injection, and Spatial Structure Control (Equations 2, 3, and 4) are all different forms of feature injection and a direct extension of feature injection [56] from image diffusion models to video diffusion models."*

**Reviewer rXMa** maintained the *weak accept* rating and had no further comments.

**Reviewer t6q7**, who initially gave an *accept* rating, flagged the confidence as less certain. After the rebuttal, **Reviewer t6q7** lowered the rating to *borderline accept*.

The paper's final ratings are, therefore, $(5, 6, 5)$. Although the final ratings are on the positive side, there is much uncertainty and concerns about the analysis in the paper.  The rebuttal presents additional helpful analysis.  However, due to the large number of submissions and limited capacity, preference must be given to papers that do not require substantial revision to be acceptable and, therefore, due to the remaining concerns and significant needed revision, the area chair unfortunately feels compelled to recommend rejecting the paper. This decision was also reviewed by the senior AC.